# Efficient Long-range Language Modeling with Self-supervised Causal Retrieval

## Abstract

Recently, retrieval-based language models (RLMs) have received much attention. However, most of them leverage a pre-trained retriever with fixed parameters, which may not adapt well to causal language models. In this work, we propose Grouped Cross-Attention, a novel module enabling joint pre-training of the retriever and causal LM, and apply it to long-context modeling. For a given input sequence, we split it into chunks and use the current chunk to retrieve past chunks for subsequent text generation. Our innovation allows the retriever to learn how to retrieve past chunks that better minimize the auto-regressive loss of subsequent tokens in an end-to-end manner. By integrating top-$k$ retrieval, our model can be pre-trained efficiently from scratch with context lengths up to 64K tokens. Our experiments demonstrate that our model achieves superior performance in various tasks against strong baselines, and 100% accuracy in the needle-in-a-haystack (NIAH) test with a 16M context length.

## 1 Introduction

Transformers (Vaswani et al., 2017), serving as the backbone of large language models (LLM), have revolutionized language modeling and demonstrated exceptional performance across a wide range of natural language processing tasks (Brown et al., 2020; Achiam et al., 2023; Touvron et al., 2023; Dubey et al., 2024). While Transformers excel in representational power, their quadratic computational complexity and increasing memory demands as input length grows pose formidable challenges for modeling long contexts. Various approaches, such as recurrent memory (Dai et al., 2019), and linear attention (Katharopoulos et al., 2020) techniques, are proposed to improve the efficiency and effectiveness of Transformers in handling extended inputs. Nevertheless, these approaches often sacrifice the random-access flexibility of attention (Mohtashami & Jaggi, 2023) during inference.

In this work, we explore long-range language modeling in Transformers from the perspective of retrieval-based language models (RLMs) (Asai et al., 2023). Typically, RLMs (Rubin & Berant, 2024; Yen et al., 2024) divide an input sequence into chunks, retrieve relevant ones from the history for the current input, and then integrate the retrieved chunks into the decoder to predict subsequent tokens. By choosing top-$k$ chunks as a "dynamic context", RLMs overcome the efficiency challenges in long-context modeling while maintaining random-access flexibility. However, most RLMs (Lewis et al., 2020; Borgeaud et al., 2022) rely on separately pre-trained retrievers with fixed parameters, which hinders their ability to adapt to the causal LMs. Although a straightforward approach is training the retriever end-to-end to select chunks that minimize auto-regressive loss of subsequent tokens, it is rarely explored. The main challenges are twofold: firstly, while relevance scores guide chunk selection, these scores do not participate in the next token prediction, thus unable to receive gradient backpropagation from the auto-regressive loss. Secondly, the large search space brought by long contexts often results in efficiency and flexibility issues for pre-training.

To tackle these challenges, we propose **G**rouped **C**ross-**A**ttention (GCA), a novel module enabling efficient end-to-end joint optimization of the retriever and causal LM, thus the retriever can *learn to retrieve* past chunks that most effectively reduce the auto-regressive loss of subsequent tokens, which we refer to as *causal retrieval*. GCA enables the relevance scores to participate in the next token prediction in a differentiable way. Specifically, GCA can be understood as a chunk-wise analogy of token-wise self-attention. In self-attention, considering the next token prediction in causal Transformers, self-attention scores could be viewed as the relevance scores of the current token to

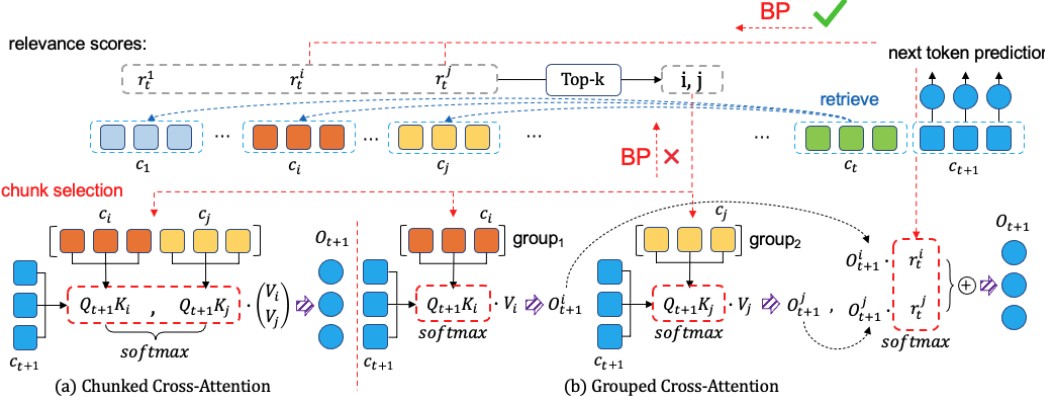

Figure 1: Comparing previous works with GCA. Consider current chunk $c_t$ with its past chunk relevance scores $r_t^k$, where $k \in \{1, \ldots, t-1\}$, $r_t^i$ and $r_t^j$ are the top two. In this example, each chunk contains 3 tokens, whose query, key, and value vectors are denoted as $Q, K, V$. (a) In previous work, information from retrieved chunks is fused into LM decoders via Chunked Cross-Attention, in which relevance scores are merely used for chunk selection. Thus the loss can not back-propagate to the scores. (b) GCA separately applies Cross-Attention with the two chunks, yielding intermediate outputs $O_{t+1}^i$ and $O_{t+1}^j$. The softmaxed relevance scores serve as weights to fuse these intermediates into LM decoders and thus can receive back-propagation from the loss.

past tokens. These scores serve as weights to fuse information gathered from past tokens to predict the next token. Analogously, we divide the input sequence into chunks and use the relevance scores between the *current* and past chunks as weights to fuse information for the *next* chunk prediction. A detailed comparison between previous works and GCA is depicted in Figure 1. By appending GCA after self-attention in Transformer layers, we introduce **D**ifferentiable **R**etrieval-based **T**ransformers (DRT), enabling pre-training from scratch with context lengths up to 64K. To make pre-training efficient, we sample top-$k$ past chunks according to the relevance scores for each chunk to perform GCA, along with fixed-size sliding window self-attention (Child et al., 2019), achieving linear complexity for the entire input sequence's attention operations. During inference, we offload hidden states of past chunks to CPU memory and reload them when retrieved. It introduces additional memory-swap but largely reduces memory footprint.

In our experiments, we evaluate our model on tasks such as long-range language modeling, summarization, and the needle-in-a-haystack (NIAH) tests. The results demonstrate that DRT significantly outperforms all baselines with comparable pre-training costs and much lower inference costs. Notably, in the NIAH test, DRT trained with a 16K context length maintains nearly 100% accuracy on inputs up to 16M tokens. More interestingly, case studies on the arXiv-math dataset suggest that long-range reasoning ability emerges in DRT, which retrieves lemmas, variants, or functions defined distantly but used in the next chunk. These findings suggest that GCA has the potential to be a fundamental component in retrieval-based LMs. Overall, our main contributions are:

1. We propose a novel module called **G**rouped **C**ross-**A**ttention (GCA), which allows dense retrievers *learn to retrieve* guided by auto-regressive loss in an end-to-end manner efficiently.

2. Building upon GCA, we introduce **D**ifferentiable **R**etrieval-based **T**ransformers (DRT), which is fast and memory-efficient in both pre-training and inference on long texts, but still maintains the random-access flexibility and excellent extrapolation capability.

3. We implement a hardware-aware GCA based on FlashAttention-2 (Dao, 2024), significantly reducing the training and inference time. The code will be made publicly available.

## 2 RELATED WORKS

**Relation to RPT & Landmark Attention.** There are two long-range LMs closely related to ours. One of them is **R**etrieval-**P**retrained **T**ransformer (RPT) (Rubin & Berant, 2024). The key difference between DRT and RPT is the training approach of the retriever. During data-preparation, for each chunk, RPT picks relevant past chunks by using BM25 (Robertson & Zaragoza, 2009), concatenates them with the current chunk, and evaluates them by a *reference LM* like Pythia 1.4B (Biderman

et al., 2023). The past chunks that increase the probability of the next chunk are identified as 'gold chunks' to train RPT's retriever. However, such a complex data preparation process limits scalability and flexibility in pre-training and post-training (Lee, 2024). In contrast, DRT is pre-trained end-to-end. By employing a sliding window size larger than the chunk size, it effectively uses feedback from subsequent several chunks to train the retriever. Its flexibility also allows for adaptive multi-hop retrieval. **L**andmark **A**ttention (LA) (Mohtashami & Jaggi, 2023) is another close work. LA is pre-trained with short contexts but capable of handling long contexts during inference. It addresses long-range language modeling by modifying self-attention KV Cache. During inference, each token, at each layer, selects top-$k$ chunks based on token-to-chunk attention scores and appends their key and value vectors to the current KV cache of self-attention. The token-to-chunk attention scores are trained in an end-to-end manner with a grouped softmax technique. However, it has to perform top-$k$ chunk selection per token, per layer, which incurs significant extra costs during inference. Moreover, it fails to extrapolate on longer context length. Our method combines the chunk-retrieval and grouped softmax ideas, resolving the aforementioned issues while balancing training efficiency and inference performance.

**Long-Range Language Modeling.** Various methods have been proposed to improve long-range language modeling. One line of research is introducing memorization to Transformers via recurrence. Many works (Dai et al., 2019; Burtsev & Sapunov, 2020; Martins et al., 2022; Hutchins et al., 2022) compress past information into fixed-sized vectors. However, these methods often sacrifice the flexibility to attend to arbitrary past tokens. Meanwhile, other works focus on maintaining random-access flexibility of attention. Memorizing Transformers (Wu et al., 2022) appends retrieved past keys and values to the current attention segment via $k$-NN search, but they do not back-propagate gradients to them. CEPE (Yen et al., 2024) retrieves previous chunks using an independently trained dense retriever and fuses them into the decoder. During the training process, the decoder parameters are fixed, and only the encoder is adjusted. A notable distinction in our work is the end-to-end optimization of all parameters, particularly the retriever.

**Efficient Language Modeling.** Many works have been done to reduce the training and inference cost of LLM. One direction is sparse attention, which includes limiting the attention window to a small range around each token (Child et al., 2019; Zaheer et al., 2020; Beltagy et al., 2020), approximating attention matrix (Wang et al., 2020), leveraging locality-sensitive-hashing(LSH) for key vectors retrieval (Kitaev et al., 2020), and hierarchical self-attention (Ren et al., 2021). However, empirically most efficient Transformers sacrifice performance for efficiency. Recently, state-space models (Gu & Dao, 2023; Dao & Gu, 2024) and RNN models (Beck et al., 2024) provide new architecture alternatives, with comparable performance to Transformers but much lower cost for inference. We argue that our core innovation GCA is flexible enough to be incorporated into these models as an additional module to obtain random-access flexibility.

**Retrieval-Augmented Language Models.** Retrieval-augmented LMs leverage a retriever to access relevant external knowledge, enhancing their generation capabilities. In some works, the retriever can be jointly trained with the LM such as REALM (Guu et al., 2020). However, its computational complexity limits its extension to causal LMs. On the other hand, in most other works (Lewis et al., 2020; Izacard & Grave, 2021; Borgeaud et al., 2022; Ivgi et al., 2023), retriever parameters are fixed, preventing optimization for retrieving information that best predicts subsequent tokens.

**Unsupervised Dense Retrieval.** In the line of research on unsupervised dense retrieval, early works leverage Inverse Cloze Task (Lee et al., 2019) to train retriever unsupervisedly, where a sentence is randomly sampled from a document and the task is to predict its surrounding context. However, this approach still lags behind BM25 on long-tail entities (Sciavolino et al., 2021). Recently, contrastive learning methods (Izacard et al., 2022; Gao & Callan, 2022) have shown improved results by creating positive and negative pairs from sentences within the same or different documents, respectively. However, these unsupervisedly trained retrievers are typically not optimized for causal LMs, so they may not guarantee to retrieve the most pertinent information.

## 3 DIFFERENTIABLE RETRIEVAL-BASED TRANSFORMER

A typical architecture of RLMs (Borgeaud et al., 2022; Yen et al., 2024; Rubin & Berant, 2024) appends Chunked Cross-Attention (CCA) after self-attention to fuse information from retrieved

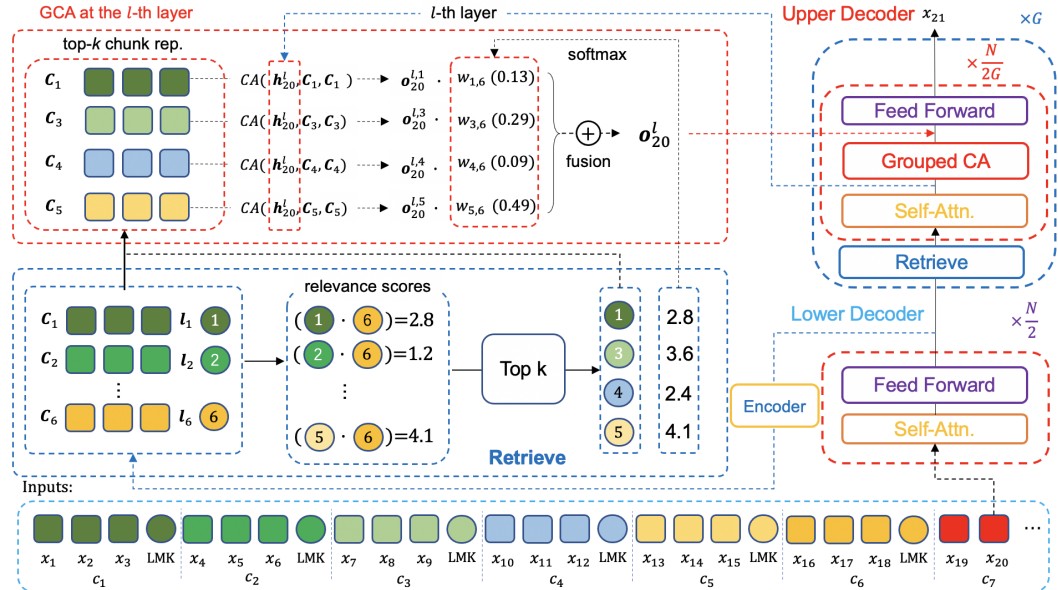

Figure 2: The illustration shows how the current chunk $c_6$ retrieves past chunks for better next token prediction of $x_{20}$ in the next chunk $c_7$. The landmark representation $\boldsymbol{l}_6$ is used to compute relevance scores with past chunks, selecting the top four. The hidden states of $x_{20}$ at the $l$-th layer, $\boldsymbol{h}_{20}^l$, perform **C**ross-**A**ttention (CA) with *all* tokens in each *separate* chunk. The chunk-wise CA outputs, $o_{20}^{l,\cdot}$, are fused via weighted sum, whose weights are softmaxed relevance scores.

chunks, in which a retriever is merely used to pick relevant chunks. Our approach makes retriever learnable by replacing CCA with GCA, which represents the key innovation of this work. The novelty of GCA lies in using relevance scores to fuse information from retrieved chunks for LM decoders, enabling the retriever to adaptively learn to select the best past chunks for predicting subsequent tokens, guided by the auto-regressive loss. This section details the model architecture, training, and inference.

## 3.1 MODEL ARCHITECTURE

DRT is composed of $N$ Transformer-like layers. Similar to RETRO (Borgeaud et al., 2022), the input sequence of DRT is equally divided into chunks. Formally, given a sequence $\mathbf{x} = [x_1, x_2, ..., x_L]$ with $L$ tokens, we divide the sequence into $\frac{L}{S}$ chunks, where $S$ is the chunk size, denoted as $\{c_1, c_2, ..., c_{L/S}\}$, where $x_i \in c_{\lceil i/S \rceil}$. Similar to Landmark Attention, we insert a special token LMK at the end of each chunk, which summarizes the preceding content via self-attention.

**Forward Pass.** Figure 2 illustrates the forward pass of a token in DRT. DRT layers are bifurcated into upper and lower sections like in RPT (Rubin & Berant, 2024). The key differences are the introduction of GCA and the further division of the upper layers into $G$ groups, enabling learning to retrieve on the fly and adaptive multi-hop retrieval. The lower layers comprise standard Transformer layers while each upper layer has an additional GCA module after self-attention. In the forward pass, the chunk hidden states output by the lower layers, besides being fed to the upper layers, are also fed into a bi-directional Transformer encoder, which further contextualizes the representations with inner-chunk positional encoding, yielding $\boldsymbol{C}_k \in \mathbb{R}^{S \times d}$ and $\boldsymbol{l}_k \in \mathbb{R}^d$ shared across all upper layers, where $d$ is the hidden state dimension. At the $g$-th upper decoder group, chunk $c_t$ retrieves the top-$k$ relevant past chunks for its next chunk:

$$r_t^{g,k} = \frac{\boldsymbol{h}_t^{g\top} \boldsymbol{l}_k}{\sqrt{d}}, \quad \mathcal{C}_t^g = \text{Top-k}([r_t^{g,1}, ..., r_t^{g,t-1}]). \tag{1}$$

Here, $\boldsymbol{h}_t^g \in \mathbb{R}^d$ represents the landmark representation output by the decoder layer just before the $g$-th group, which accumulates all information from groups 1 to $g - 1$. This enables $\boldsymbol{h}_t^g$ to retrieve relevant information based on previously retrieved chunks thus achieving multi-hop retrieval. $r_t^{g,k}$ represents the causal relevance score of $c_k$ to $c_t$. $\mathcal{C}_t^g$ contains the indices of past chunks with top-$k$

relevance scores. The retrieved chunks are shared among the subsequent layers within the same group. The upper layers apply GCA to fuse retrieved information into the decoder.

**Grouped Cross-Attention.** For the $l$-th layer, let $\boldsymbol{H}_{t+1}^l$ and $\hat{\boldsymbol{H}}_{t+1}^l \in \mathbb{R}^{(S+1)\times d}$ denote the batched token representations including the landmark token in the next chunk before and after GCA. In GCA, we perform Cross-Attention (CA) separately and fuse results via relevance scores:

$$g(l) = \lceil (l - \frac{N}{2})/\frac{N}{2G} \rceil, \quad \boldsymbol{O}_{t+1,k}^l = \mathbf{CA}(\boldsymbol{H}_{t+1}^l, \boldsymbol{C}_k, \boldsymbol{C}_k), \quad k \in \mathcal{C}_t^{g(l)},$$

$$w_t^{g(l),k} = \frac{\exp(r_t^{g(l),k})}{\sum_{k' \in \mathcal{C}_t^{g(l)}} \exp(r_t^{g(l),k'})}, \boldsymbol{O}_{t+1}^l = \sum_k w_t^{g(l),k} \boldsymbol{O}_{t+1,k}^l, \quad \hat{\boldsymbol{H}}_{t+1}^l = \mathrm{Norm}(\boldsymbol{H}_{t+1}^l + \boldsymbol{O}_{t+1}^l). \tag{2}$$

Here $g(l)$ converts the layer index to the group index and $\boldsymbol{C}_k \in \mathbb{R}^{S\times d}$ represents token representations of the $k$-th retrieved chunk. $\boldsymbol{O}_{t+1,k}^l \in \mathbb{R}^{(S+1)\times d}$ represents the information that $S+1$ tokens in chunk $c_{t+1}$ gather from past chunk $c_k$. $w_t^{g(l),k}$ is the normalized relevance score after softmax, serving as the weight of $\boldsymbol{O}_{t+1,k}^l$ for information fusion. The final fused results of GCA is $\boldsymbol{O}_{t+1}^l$.

Since $\boldsymbol{C}_k$ is shared across layers, we use the same K, V liner transformations across layers to compact model parameters and reduce memory footprint. For each head $h$, we have CA defined as:

$$\mathbf{CA}(\boldsymbol{H}_{t+1}^l, \boldsymbol{C}_k, \boldsymbol{C}_k)_h \triangleq \mathrm{Softmax}(\frac{Q_h^l(\boldsymbol{H}_{t+1}^l)K_h(\boldsymbol{C}_k)^T}{\sqrt{d_h}})V_h(\boldsymbol{C}_k) \tag{3}$$

Here, $d_h$ is per head dimension, and $Q_h^l, K_h, V_h$ are linear transformations per head, where $Q_h^l$ varies across layers and $K_h, V_h$ are layer-shared. The final CA outputs are concatenated vectors from all heads. It is worth noting that GCA is easy to integrate with FlashAttention-2, as detailed in the pseudo-code in Appendix A.2.

**GCA vs CCA.** A key distinction between GCA and CCA lies in how softmax is applied to cross-attention matrices as shown in Figure 1(a)(b). In CCA, all retrieved chunks are concatenated and softmax is directly applied to the whole attention matrix to fuse token-level information. Notably, relevance scores are entirely excluded from the process. In contrast, GCA applies softmax to each chunk's attention matrix separately. This modification allows information to be gathered from each chunk separately. The softmaxed relevance scores then serve as soft choices (Hu et al., 2021; 2022), participating in the next token prediction thus receiving back-propagated gradients.

**Encoder-Decoder Variant.** Our model architecture could be easily modified to an encoder-decoder-based variant like RETRO (Borgeaud et al., 2022) by directly applying the encoder to chunk token embeddings instead of the lower layers' outputs. This variant allows for retrieval from trillions of tokens with acceptable storage costs. We will elaborate on this in § 3.3.

## 3.2 TRAINING

The pre-training objective of DRT is the next token prediction, but there are certain details as discussed below. We also give a detailed time complexity analysis of training.

**Gumbel Top-k sampling.** The core idea of self-supervised retrieval is making candidate chunks compete with each other by softmaxing relevance scores as weights. The weights of the chunks contributing most to the next chunk prediction are enhanced while the weights of the rest are suppressed. To balance exploration and exploitation, we sample chunks based on relevance scores instead of always picking the top-k, enabling highly relevant chunks to be more likely chosen while still considering lower-scoring ones. A simple trick is to add Gumbel noise (Gumbel, 1954) to the raw scores before the top-k operation. Importantly, this noise doesn't affect subsequent operations.

**Encoder-Decoder Pre-training.** To enhance the encoder's representational ability, we train it with MLM in addition to the auto-regressive loss during the first half of pre-training. Specifically, while the decoder sees all tokens, the encoder's tokens are partially masked, whose outputs are used in both the decoder's GCA and computing the MLM objective. Masking causes inaccurate encoding that may disturb subsequent computation and training of the decoder, so we eventually remove the MLM training in the second half of the pretraining.

**Time Complexity.** Our approach reduces training complexity by compressing quadratic operations. Vanilla Transformers have a complexity of $\mathcal{O}(NL^2)$ for full self-attention. In DRT, we encode chunks with $S$ tokens into landmark representations, performing chunk-wise full attention to compute relevance scores within $\mathcal{O}(G\frac{L^2}{S^2})$ for $G$ groups of upper layers. By employing sliding-window attention and top-$k$ retrieval, we scale down self-attention and GCA costs to $\mathcal{O}(G\frac{L^2}{S^2} + \frac{N}{2}LKS + NLW)$, where $K$ is the retrieved chunk number and $W$ is the window size, $K, W \ll L$. This largely reduces the complexity but maintains the random-access flexibility.

## 3.3 Inference

**Memory Offloading.** During the inference stage of Transformers, the default memory cost for KV Cache is $\mathcal{O}(NLd)$. To reduce GPU memory usage, we can offload past chunk representations to CPU memory. This results in a spatial complexity of the GPU memory usage $\mathcal{O}(\frac{Ld}{S} + \frac{N}{2}KSd + NWd)$ during inference. Here, $\frac{Ld}{S}$ is the memory footprint of landmark representations, while the remaining terms account for the GCA and sliding-window KV cache. Although each retrieval involves gathering representations from CPU memory and transferring them to the GPU, this operation occurs $G$ times every $S$ tokens. Therefore, the cost of memory exchange from chunk retrievals is minimal.

**Infinitely Long Context Retrieval.** To mimic humans' capability for random access to memories, we extend the retrievable context to all pre-trained tokens, where contextualized token representations in chunks could be regarded as memory. A straightforward approach, similar to RPT, is to store chunk-level representations and token-level hidden states as key-value pairs in a Faiss (Douze et al., 2024). However, this approach requires significant disk space. For instance, a 100 billion token corpus with 1,024-dimensional hidden states, even with Product Quantization, demands approximately 25TB. In contrast, using an encoder-decoder variant, we only store a chunk's landmark representation and positions in the corpus without saving hidden states. When retrieving, we access the corresponding tokens by document position and re-encode them to obtain their hidden states, achieving retrieval from billions of tokens with disk cost reducing by $S$ fold.

## 4 Experiments

We compare DRT with prior works in long-range language modeling, wall-clock training time, inference cost, extrapolation capability, and infinite-length context retrieval. Notably, we include the closely related works RPT and Landmark Attention.

### 4.1 Experimental Setup

#### 4.1.1 Datasets

**PG19.** PG19 (Rae et al., 2020) is a language modeling benchmark widely used to evaluate long-range text understanding capabilities of models. It includes a collection of full-length books from Project Gutenberg across a wide range of genres, styles, and topics. The dataset is particularly useful for evaluating models' abilities to capture dependencies and context over long sequences of text.

**ArXiv-math.** ArXiv-math is a corpus consisting of mathematical papers from arXiv. Characterized by a sustained coherence over long distance, the corpus requires models' capability of correctly referring to long-range history information and using long-range context effectively for predictions. We use the preprocessed corpus and data splits from Azerbayev et al. (2023).

**MiniPile.** MiniPile (Kaddour, 2023) is a 6GB subset of the deduplicated 825GB *The Pile* (Gao et al., 2021) corpus, which covers sub-datasets from webpages, dialogue, books, science and code.

#### 4.1.2 Models

**DRT$_{\text{retrieval} \times G}$.** A DRT consists of 12 Transformer decoder layers, divided into 6 lower and 6 upper layers, with upper layers further split into $G$ groups. The sliding window size is set to $W$=512, the chunk size is set to $S$=64, and 8 chunks are retrieved for GCA, resulting in an attention field of

512 ($8 \times 64$). We implement hardware-aware GCA based on Triton (Tillet et al., 2019) [1]. As we employ various parameter settings across different experiments, further details are provided in Appendix A.1. **DRT$_{\text{enc-dec}}$:** the variant of DRT introduced in § 3.1.

**Base LM.** Our base LM is based on the implementation of TinyLlama (Zhang et al., 2024) combined with Flash Attention2 (Dao, 2024) enabling ALiBi (Press et al., 2022) and sliding window attention (Child et al., 2019). We compare models against the baseline across various configurations. One configuration involves 12 layers with a sliding window of 512 tokens, aligning with the DRT sliding window size. Another configuration of 12 layers with a 768-token sliding window ensures the same attention field coverage, as $12 \times 768 = 12 \times 512 + 6 \times 512(\text{GCA})$. The strongest baseline, with 14 layers and a 658-token sliding window, has a parameter count comparable to our DRT while maintaining a similar total attention field across all 12 layers, calculated as $658 \times 14 \approx 12 \times 512 + 6 \times 512$.

**Retrieval-Pretrained Transformer (RPT).** Since the official implementation is in JAX and the code for distilling the retriever is not released, we reimplement RPT in PyTorch and replace the retriever with Contriever (Izacard et al., 2022). Elaborations can be found in Appendix A.6.

**Landmark Attn.** We use the official Llama-like implementation [2] of Landmark Attention. Similar to Base LM, we extend the length of the self-attention range from 512 to 768 to ensure it shares the same attention field as DRT.

**Block-Recurrent TFM.** Since the official implementation of Block-Recurrent Transformer is also based on JAX, we utilized a PyTorch implementation [3] to ensure all baselines are running with the same framework.

**Ablations.** *w/o Triton*: A naively implemented version of GCA without Triton. *w/o enc*: The decoder-only version of DRT, which uses the hidden states of the middle layer of the decoder stack to retrieve history chunks. Differently put, this model can be attained by simply removing the encoder from DRT$_{\text{enc-dec}}$. *w/o mlm*: The architecture is exactly the same as DRT$_{\text{enc-dec}}$, while the only difference lies in the training process by eliminating the masked language modeling part. *w/o gumbel top-k*: The architecture is exactly the same as DRT, while the only difference lies in the training process by eliminating the gumbel noise when selecting the top-k chunks.

## 4.2 Long-range Language Modeling

In this section, we evaluate DRT against baselines in long-range language modeling on PG19 and arXiv-math, and report their respective perplexities. All models are pre-trained with the same attention field and a 16K context by default, except for baselines that cannot efficiently pre-train on long contexts. To ensure fairness, we adjusted these baselines. Detailed hyper-parameters are provided in Appendix A.1.

**Results.** From Table 1, we have several observations. **Firstly**, DRT outperforms all baselines where the evaluation length exceeds 16K. While DRT performs retrieval for $G$ times every 64 tokens, LA performs retrieval at every token and in every layer, potentially offering better random access flexibility. However, DRT still surpasses LA on longer inputs. This is likely because LA follows a "train short, test long" approach due to the need for full attention during pre-training, whereas DRT can be directly pre-trained on long input sequences. Thanks to GCA, our key innovation, DRT can randomly access distant contexts during pre-training with the same attention field as in the baselines, allowing it to better utilize long-range information during pre-training with negligible extra training costs. **Secondly**, in terms of parameter efficiency, we compared against Base LM with two additional layers in the decoder stack. It can be observed that on shorter evaluation lengths, the baseline has a slight advantage. However, with a longer context, our model consistently leads. This experiment suggests precisely retrieving semantic knowledge from a long context may be more beneficial for improving the language model than increasing model parameters. **Thirdly**, multi-hop retrieval yields positive gains with a low marginal cost. It allows upper layers to access more diverse

---

[1] https://github.com/triton-lang/triton/blob/main/python/tutorials/06-fused-attention.py

[2] https://github.com/epfml/landmark-attention

[3] https://github.com/lucidrains/block-recurrent-transformer-pytorch

| Model | Time | #Param. dec/enc | k | attn. win. | PG19↓ valid | PG19↓ test | ArXiv-math↓ valid | ArXiv-math↓ test |
|---|---|---|---|---|---|---|---|---|
| Train length=16K, eval length = 2K | | | | | | | | |
| BaseLM | 1× | 124M | - | 512 | 15.00 | 14.10 | 3.31 | 3.31 |
| (Sliding window+Alibi) | 1.03× | 124M | - | 768 | 14.86 | 13.96 | 3.24 | 3.24 |
| +2 layers | 1.15× | 138M | - | 658 | 14.71 | 13.83 | **3.06** | **3.06** |
| RPT$_{contriever}$(our impl.) | 2.5× | 133M/14M | 8 | 512 | 14.81 | 13.92 | 3.24 | 3.24 |
| Landmark Attn. | 1.5× | 124M | 4 | 768 | **14.41** | **13.40** | 3.17 | 3.16 |
| Block Recurrent TFM | 2× | 155M | - | 768 | 15.99 | 15.00 | 3.33 | 3.32 |
| DRT$_{enc-dec}$ | 1.22× | 133M/14M | 4 | 512 | 14.90 | 14.02 | 3.24 | 3.24 |
| DRT$_{retrieval×1}$ | 1.22× | 133M/14M | 8 | 512 | 14.65 | 13.78 | 3.24 | 3.24 |
| DRT$_{retrieval×2}$ | 1.24× | 133M/14M | 8 | 512 | 14.56 | 13.69 | 3.22 | 3.22 |
| Train length=16K, eval length = 16k | | | | | | | | |
| BaseLM | 1× | 124M | - | 512 | 14.55 | 13.68 | 3.06 | 3.06 |
| (Sliding window+Alibi) | 1.03× | 124M | - | 768 | 14.36 | 13.49 | 2.95 | 2.95 |
| +2 layers | 1.15× | 138M | - | 658 | 14.23 | 13.37 | 2.95 | 2.94 |
| RPT$_{contriever}$(our impl.) | 2.5× | 133M/14M | 8 | 512 | 14.39 | 13.52 | 2.93 | 2.92 |
| Landmark Attn. | 1.5× | 124M | 4 | 768 | 14.10 | 13.21 | 3.02 | 3.02 |
| Block Recurrent TFM | 2× | 155M | - | 768 | 15.59 | 14.60 | 3.14 | 3.14 |
| DRT$_{enc-dec}$ | 1.22× | 133M/14M | 4 | 512 | 14.42 | 13.53 | 2.93 | 2.93 |
| DRT$_{retrieval×1}$ | 1.22× | 133M/14M | 8 | 512 | 14.05 | 13.21 | 2.89 | 2.89 |
| DRT$_{retrieval×2}$ | 1.24× | 133M/14M | 8 | 512 | **14.02** | **13.18** | **2.85** | **2.85** |
| Train length=16K, eval length = 32k | | | | | | | | |
| BaseLM | 1× | 124M | - | 512 | 14.50 | 13.64 | 3.05 | 3.04 |
| (Sliding window+Alibi) | 1.03× | 124M | - | 768 | 14.30 | 13.46 | 2.93 | 2.92 |
| +2 layers | 1.15× | 138M | - | 658 | 14.18 | 13.34 | 2.93 | 2.92 |
| RPT$_{contriever}$(our impl.) | 2.5× | 133M/14M | 8 | 512 | 14.35 | 13.49 | 2.91 | 2.91 |
| Landmark Attn. | 1.5× | 124M | 4 | 768 | 14.19 | 13.33 | 3.07 | 3.07 |
| Block Recurrent TFM | 2× | 155M | - | 768 | 15.61 | 14.56 | 3.13 | 3.12 |
| DRT$_{enc-dec}$ | 1.22× | 133M/14M | 4 | 512 | 14.38 | 13.52 | 2.91 | 2.91 |
| DRT$_{retrieval×1}$ | 1.22× | 133M/14M | 8 | 512 | 14.01 | 13.19 | 2.85 | 2.85 |
| DRT$_{retrieval×2}$ | 1.24× | 133M/14M | 8 | 512 | **13.98** | **13.16** | **2.81** | **2.81** |
| Ablation studies | | | | | | | | |
| –w/o Triton | 1.45× | 133M/14M | 8 | 512 | — | — | — | — |
| –w/o enc. | -eval len=16k | | 8 | 512 | 14.31 | 13.44 | 2.97 | 2.97 |
| –w/o MLM | -eval len=16k | | 4 | 512 | 14.43 | 13.57 | 3.07 | 3.06 |
| –w/o gumbel top-k | -eval len=16k | | 8 | 512 | 14.36 | 13.46 | 2.90 | 2.90 |

Table 1: Perplexity for all datasets. We highlight the best results in **bold** and underline the second best.

chunks and enables further retrieval based on previous retrieval results. **Finally**, ablation studies show that all the training techniques we add bring positive improvements. In conclusion, the above results fully demonstrate that the GCA module can indeed bring effective gains in modeling long texts, and it is more advantageous compared to other baselines.

## 4.3 DOWNSTREAM TASK EVALUATION AND EFFICIENCY ANALYSIS

In this section, we fine-tune all baselines and evaluate them against downstream tasks including summarization (Nallapati et al., 2016; Narayan et al., 2018), single NIAH test, and multi-hop NIAH proposed by Hsieh et al. (2024). The details for the downstream tasks are described in Appendix A.4. Then we analyze the inference cost, the relationship between training time and context length, and the extrapolation capability of DRT. In the inference cost analysis, We skip RPT because it has a similar cost to DRT. In the extrapolation experiments, we utilize the pre-trained models described in § 4.2 to assess their performance with extended context lengths.

**Results.** From Table 2, we observe that DRT significantly outperforms all baselines in the summarization tasks, validating its capability to effectively utilize long contexts. Notably, in the single NIAH test, DRT maintains 100% performance even with a context length of up to 16 million tokens, demonstrating its strong length generalization ability in long-context scenarios. Furthermore, in the 2-hop NIAH test, DRT$_{retrieval×2}$ performs comparably to Landmark Attention at context lengths below 16K tokens while successfully extrapolating to longer context lengths beyond 64K tokens. Additionally, DRT$_{retrieval×2}$ significantly outperforms DRT$_{retrieval×1}$, confirming our hypothesis that conducting causal retrieval every $G$-layer contributes to scenarios requiring multi-hop retrievals.

| Models | Retrieval | Single NIAH↑ | | | | | | | |
|---|---|---|---|---|---|---|---|---|---|
| | | 1K | 2K | 4K | 8K | 16K | 32K | 64K | 16M |
| Base LM (+2 layers) | – | 60.43 | 29.54 | 15.37 | 8.30 | 3.89 | 2.13 | 0.0 | - |
| Block Recurrent TFM | – | 66.80 | 30.61 | 13.96 | 7.60 | 6.01 | 2.13 | 4.29 | - |
| RPT$_{Contriever}$ | fixed | 42.13 | 18.45 | 11.66 | 6.71 | 4.24 | 1.42 | 2.86 | - |
| Landmark Attn. | adaptive | **99.98** | 99.08 | **99.82** | 97.74 | 97.88 | 96.45 | 0.00 | - |
| DRT$_{retrieval \times 1}$ | adaptive | 98.23 | 99.12 | 98.50 | 98.76 | 98.59 | **100.00** | **100.00** | **100.00** |
| DRT$_{retrieval \times 2}$ | adaptive | 99.69 | **99.56** | 99.65 | **99.47** | **99.65** | 99.99 | **100.00** | **100.00** |

| Models | XSum↑ | | | CNN/DailyMail↑ | | | 2-hop NIAH with noises↑ | | | | | |
|---|---|---|---|---|---|---|---|---|---|---|---|---|
| | R-1 | R-2 | R-L | R-1 | R-2 | R-L | 1K | 4K | 16K | 64K | 256K | 4M |
| BaseLM | 29.43 | 8.04 | 23.26 | 32.38 | 12.57 | 22.61 | 3.60 | 1.15 | 0.71 | 0.0 | 0.0 | - |
| +2 layers | 29.74 | 8.26 | 23.48 | 34.14 | 14.09 | 23.60 | 16.60 | 5.83 | 1.06 | 0.0 | 0.0 | - |
| Landmark Attn. | 27.98 | 6.96 | 21.99 | 34.06 | 13.80 | 23.77 | **90.82** | **88.35** | **86.41** | 0.0 | 0.0 | - |
| DRT$_{retrieval \times 1}$ | 30.30 | 8.59 | 23.92 | 36.27 | 15.88 | 25.08 | 41.07 | 33.39 | 39.93 | 38.57 | 35.29 | 34.29 |
| DRT$_{retrieval \times 2}$ | **30.39** | **8.64** | **23.98** | **36.39** | **15.96** | **25.15** | 88.52 | 84.45 | 86.21 | **81.43** | **94.11** | **79.41** |

Table 2: The performances of various models on summarization tasks and NIAH tests.

| | Prompt #tokens | Generated #tokens | w/ cpu offload | | w/o cpu offload | |
|---|---|---|---|---|---|---|
| | | | time/token↓ | mem. cost↓ | time/token↓ | mem. cost↓ |
| Landmark Attn. / Base LM | 16K | 128 | 160.9× | 1.98× | 4.16× | 32× |
| | 48K | 128 | 163.1× | 2.98× | 4.25× | 96× |
| Block Recurrent TFM / Base LM | 16K | 128 | - | - | 2.85× | 2× |
| | 48K | 128 | - | - | 2.85× | 2× |
| DRT$_{retrieval \times 1}$ / Base LM | 16K | 128 | 1.25× | 1.54× | 1.06× | 4.08× |
| | 48K | 128 | 1.27× | 1.62× | 1.08× | 9.41× |

Table 3: The inference time per token and memory footprint ratio compared to the Base LM (12 layers with a 512 sliding window), with lower values indicating better performance.

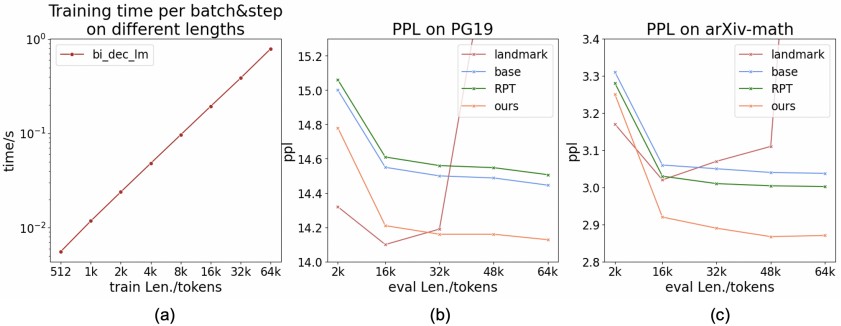

(a)  (b)  (c)

Figure 3: Training speed, extrapolation ability

Table 3 shows DRT significantly outperforms Landmark Attn in terms of memory footprint and inference speed. The main overhead for Landmark Attn arises from modifications to the self-attention KV cache and gathering tensors from memory offloaded to the CPU. In DRT, thanks to the chunk-wise retrieval, we perform retrieval only once every 64 tokens, which is $1/(12 \times 64)$ of the corresponding operation in Landmark Attn.

From Figure 3(a), it can be observed that the total training time increases linearly with the increment of the sequence length. In the extrapolation experiments, both BaseLM and DRT perform well as shown in Figure 3(b)(c). However, Landmark Attn fails to extrapolate at longer eval lengths. We believe a possible explanation is that a longer context increases the probability of retrieving irrelevant distant chunks, which stems from its limitations in pre-training directly on long contexts. An overly short pre-training window prevents access to longer-range information, making the model unable to learn to reduce the attention scores of long-range noise chunks. DRT benefits from being able to pre-train directly on long contexts, alleviating this issue to a certain extent.

### 4.4 RETRIEVAL FROM SIMULATED INFINITELY LONG CONTEXT

We wonder whether a densely pre-trained retriever can generalize from a limited to an unlimited context. To verify this, we simulate an infinitely long context by retrieving from all pre-trained tokens. The details could be found in Appendix A.5.

| Model | Corpus Retrieval | Self Retrieval | MiniPile↓ |
|---|---|---|---|
| Base LM (attn. win. 512) | No | No | 12.68 |
| DRT w/ random retrieval | Yes | No | 12.68 |
| DRT w/ Contriever | Yes | No | 12.65 |
| DRT w/ Contriever | No | Yes | 12.25 |
| DRT | Yes | No | 12.67 |
| DRT | Yes | Yes | 12.30 |
| DRT | No | Yes | 12.18 |

**Settings:**

*Self-Retrieval*:
Indicates if retrieval of past 48K tokens is enabled.

*Corpus-Retrieval*:
Indicates if retrieval from the pre-training corpus is enabled.

Table 4: Valid set perplexity for MiniPile under different settings.

**Results.** From Table 4, we observe that the perplexity of DRT slightly rises instead of declines when we extend the context to infinity. Retrieving information from a limited-length context yielded the best results among all methods. A possible explanation is that retrieval from the vast amount of chunks from the corpus may yield results with similar representations but semantically irrelevant content. Notably, only Contriever benefits from corpus-retrieval when self-retrieval is disabled. The key distinction between Contriever and DRT's inherent retriever is that Contriever incorporates random negative sampling during training. However, unlike Contriever, our negative samples are drawn from a fixed-size context, meaning the negative sample candidates are fixed. Contriever, on the other hand, performs random negative sampling via in-batch sampling, allowing for a more extensive negative sample space. This insight could potentially contribute to retrieval from trillion tokens in future works.

## 4.5 CASE STUDIES

*... We start first with the following* lemma which is useful to establish the Quillen equivalence. \begin{lemma} \label{lem-reflect-equiv} Let $\mathbf{M}$ be a symmetric monoidal model category that is combinatorial and left proper. Assume that the transferred (natural) model structure on *com* exists. *Let* $\sigma : \mathbf{C} \to \mathbf{D}$ *be a morphism between usual commutative...*

*... weak equivalence between co-Segal categories is just a level-wise weak equivalence.* \begin{prop} \label{prop-eta-kx-loc-equiv} For any $\mathbf{F} \in coms$, the canonical map $\mathbf{F} \to |\mathbf{F}|$ is an equivalence in $comsepc$ i.e, it's a $kb(I)$-local equivalence in $comsep$ *(whence in comse)*. \end{prop}...

*... Thanks to* **Proposition \ref{prop-eta-kx-loc-equiv}**, **we know that** $\eta : \mathbf{F} \to |\mathbf{F}|$ **is always a** $kb(I)$**-local equivalence. Then by** 3**-for-**2 we see that $\sigma$ is a $kb(I)$-local equivalence if and only if $ol(\sigma)$ is. Now thanks to Lemma \ref{lem-reflect-equiv} we know that ...

Figure 4: In the case above, Retrieved 2nd best chunk describes a proposition which appears in the **current chunk**. retrieved best chunk and retrieved 3rd best chunk are adjacent chunks which introduce the lemma used in the next chunk.

By analyzing DRT's retrieval results on the arXiv-math dataset, we find some intriguing cases. A case is given in Figure 4. When retrieving past chunks, the results not only include the definition of prepositions referenced in the current chunk but also lemmas to be used in the next chunk. This validates the idea of causal retrieval, allowing us to not only retrieve semantically similar content but also information that better predicts the next chunk. More case studies can be found in Appendix A.3.

## 5 CONCLUSION & FUTURE WORKS

In this study, we successfully optimize the retriever module with the auto-regressive LM objective in an end-to-end manner. The core innovation lies in the Grouped Cross-Attention (GCA), which makes relevance scores learnable by using them to fuse information retrieved by the current chunk for next chunk prediction. Combined with Gumbel top-k sampling, this approach enables the pre-training of LMs on context lengths extending up to 64K tokens.

In future work, we will explore self-supervised causal retrieval from vast amounts of tokens outside the context. Meanwhile, we will combine structured representations (Hu et al., 2024b;a) to achieve multi-granular retrieval.

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

# A    APPENDIX

## A.1    HYPER-PARAMETERS

**Long-Range Language Modeling.**    We employ a Llama-like architecture (Touvron et al., 2023) featuring a 12-layer, decoder-only transformer with 12 heads per layer (64 dimensions each), an embedding dimension of 768, and an FFN size of 2048. Training utilizes the AdamW optimizer (Loshchilov & Hutter, 2019) with $\beta_1 = 0.9$ and $\beta_2 = 0.95$, and a weight decay factor of 0.001. We used base learning rate $2 \times 10^{-3}$ for all our experiments with a warmup stage that was 2% of the whole training and applied a cosine scheduler with final learning rate being $4 \times 10^{-4}$. We used GPT-2's (Radford et al., 2019) tokenizer. We used mixed-precision training with bfloat16 over at 8 Nvidia A100 GPUs. We train all models with an effective batch size of $2^{19}$ tokens for 60K steps resulting in a total training budget of 32.2 billion tokens. We train Base LM, RPT and DRT on each dataset with a context length of 16K tokens. Due to Landmark Attention doesn't support sliding-window attention, the model is pre-trianed with full self-attention with a context length of 768. Due to Block Recurrent Transformer cannot be fully paralleled, which takes $5\times$ wall-clock training time with 16K context length, we pre-train it with a context length of 4K.

## A.2    HARDWARE-AWARE GCA PSUEDO-CODE

---

**Algorithm 1** FLASHGCA forward pass

---

**Require:** Matrices $\mathbf{Q} \in \mathbb{R}^{N_q \times d}, \mathbf{K}, \mathbf{V} \in \mathbb{R}^{K \times N_{kv} \times d}$ in HBM, vector $\boldsymbol{w} \in \mathbb{R}^k$ in HBM, block sizes $B_c, B_r$.

1: Divide $\mathbf{Q}$ into $T_r = \left\lceil \frac{N_q}{B_r} \right\rceil$ blocks $\mathbf{Q}_1, \ldots, \mathbf{Q}_{T_r}$ of size $B_r \times d$ each, and divide $\mathbf{K}, \mathbf{V}$ in to $K \times T_c$ blocks

   where $T_c = \left\lceil \frac{N_{kv}}{B_c} \right\rceil$ $\mathbf{K}_{1,1}, \ldots, \mathbf{K}_{K,T_c}$ and $\mathbf{V}_{1,1}, \ldots, \mathbf{V}_{K,T_c}$, of size $B_c \times d$ each.

2: Divide the output $\mathbf{O} \in \mathbb{R}^{N_q \times d}$ into $T_r$ blocks $\mathbf{O}_i, \ldots, \mathbf{O}_{T_r}$ of size $B_r \times d$ each, and divide the logsumexp $L \in \mathbb{R}^{N_q \times K}$ into $T_r \times K$ blocks $L_{1,1}, \ldots, L_{T_r,K}$ of size $B_r$ each.

3: Divide the output $\mathbf{O}' \in \mathbb{R}^{K \times N_q \times d}$ into $T_r$ blocks $\mathbf{O}_{1,1}, \ldots, \mathbf{O}_{K,T_r}$ of size $K \times B_r \times d$ each.

4: **for** $1 \leq i \leq T_r$ **do**

5:    Load $\mathbf{Q}_i$ from HBM to on-chip SRAM.

6:    Load $\boldsymbol{w}_k$ from HBM to on-chip SRAM.

7:    **for** $1 \leq k \leq K$ **do**

8:      On chip, initialize $\mathbf{O}_i^{(0)} = (0)_{B_r \times d} \in \mathbb{R}^{B_r \times d}, \ell_i^{(0)} = (0)_{B_r} \in \mathbb{R}^{B_r}, m_i^{(0)} = (-\infty)_{B_r} \in \mathbb{R}^{B_r}$.

9:      **for** $1 \leq j \leq T_c$ **do**

10:        Load $\mathbf{K}_{k,j}, \mathbf{V}_{k,j}$ from HBM to on-chip SRAM.

11:        On chip, compute $\mathbf{S}_i^{(j)} = \mathbf{Q}_i \mathbf{K}_{k,j}^T \in \mathbb{R}^{B_r \times B_c}$.

12:        On chip, compute $m_i^{(j)} = \max(m_i^{(j-1)}, \mathrm{rowmax}(\mathbf{S}_i^{(j)})) \in \mathbb{R}^{B_r}, \tilde{\mathbf{P}}_i^{(j)} = \exp(\mathbf{S}_i^{(j)} - m_i^{(j)}) \in \mathbb{R}^{B_r \times B_c}$ (pointwise), $\ell_i^{(j)} = e^{m_i^{j-1} - m_i^{(j)}} \ell_i^{(j-1)} + \mathrm{rowsum}(\tilde{\mathbf{P}}_i^{(j)}) \in \mathbb{R}^{B_r}$.

13:        On chip, compute $\mathbf{O}_i^{(j)} = \mathrm{diag}(e^{m_i^{(j-1)} - m_i^{(j)}})^{-1} \mathbf{O}_i^{(j-1)} + \tilde{\mathbf{P}}_i^{(j)} \mathbf{V}_{k,j}$.

14:      **end for**

15:      On chip, compute $\mathbf{O}'_{i,k} = \mathrm{diag}(\ell_i^{(T_c)})^{-1} \mathbf{O}_i^{(T_c)}$.

16:      On chip, compute $\mathbf{O}_i \leftarrow \mathbf{O}_i + \boldsymbol{w}_k \mathbf{O}'_{i,k}$.

17:      Write $O_{i,k}$ to HBM.

18:      On chip, compute $L_{i,k} = m_i^{(T_c)} + \log(\ell_i^{(T_c)})$.

19:      Write $L_{i,k}$ to HBM.

20:    **end for**

21:    Write $\mathbf{O}_i$ to HBM as the $i$-th block of $\mathbf{O}$.

22: **end for**

23: Return the output $\mathbf{O}$ and the logsumexp $L$.

---

**Algorithm 2** FLASHGCA Backward Pass

**Require:** Matrices $\mathbf{Q}, \mathbf{O}, \mathbf{dO} \in \mathbb{R}^{N_q \times d}, \mathbf{K}, \mathbf{V} \in \mathbb{R}^{K \times N_{kv} \times d}, L \in \mathbb{R}^{N_q \times K}, \mathbf{O}' \in \mathbb{R}^{K \times N_q \times d}$ in HBM, vector $\boldsymbol{w} \in \mathbb{R}^K$ in HBM, block sizes $B_c, B_r$.

1: Divide $\mathbf{Q}$ into $T_r = \left\lceil \frac{N}{B_r} \right\rceil$ blocks $\mathbf{Q}_1, \ldots, \mathbf{Q}_{T_r}$ of size $B_r \times d$ each, and divide $\mathbf{K}, \mathbf{V}$ in to $K \times T_c$, where
$T_c = \left\lceil \frac{N}{B_c} \right\rceil$ blocks $\mathbf{K}_{1,1}, \ldots, \mathbf{K}_{K,T_c}$ and $\mathbf{V}_{1,1}, \ldots, \mathbf{V}_{K,T_c}$, of size $B_c \times d$ each.

2: Divide $\mathbf{O}$ into $T_r$ blocks $\mathbf{O}_i, \ldots, \mathbf{O}_{T_r}$ of size $B_r \times d$ each, divide $\mathbf{dO}$ into $T_r$ blocks $\mathbf{dO}_i, \ldots, \mathbf{dO}_{T_r}$ of size $B_r \times d$ each, and divide $L$ into $T_r \times K$ blocks $L_{1,1}, \ldots, L_{T_r,K}$ of size $B_r$ each.

3: Initialize $\mathbf{dQ} = (0)_{N_q \times d}$ in HBM and divide it into $T_r$ blocks $\mathbf{dQ}_1, \ldots, \mathbf{dQ}_{T_r}$ of size $B_r \times d$ each. Divide $\mathbf{dK}, \mathbf{dV} \in \mathbb{R}^{K \times N_{kv} \times d}$ in to $K \times T_c$ blocks $\mathbf{dK}_{1,1}, \ldots, \mathbf{dK}_{K,T_c}$ and $\mathbf{dV}_{1,1}, \ldots, \mathbf{dV}_{K,T_c}$, of size $B_c \times d$ each. Initialize $\boldsymbol{dW} = (0)_{T_r \times K}$ in HBM.

4: Compute $D = \mathrm{rowsum}(\mathbf{dO} \circ \mathbf{O}') \in \mathbb{R}^{N_q \times K}$ (pointwise multiply), write $D$ to HBM and divide it into $T_r$ blocks $D_1, \ldots, D_{T_r}$ of size $B_r$ each.

5: **for** $1 \le k \le K$ **do**

6:     Load $\boldsymbol{w}_k$ from HBM to on-chip SRAM.

7:     **for** $1 \le j \le T_c$ **do**

8:         Load $\mathbf{K}_{k,j}, \mathbf{V}_{k,j}$ from HBM to on-chip SRAM.

9:         Initialize $\mathbf{dK}_{k,j} = (0)_{B_c \times d}, \mathbf{dV}_{k,j} = (0)_{B_c \times d}, \boldsymbol{dW}_{k,j} = (0)$ on SRAM.

10:         **for** $1 \le i \le T_r$ **do**

11:             Load $\mathbf{Q}_i, \mathbf{dO}_i, \mathbf{dQ}_i, D_i$ from HBM to on-chip SRAM.

12:             Load $L_{i,k}$ from HBM to on-chip SRAM.

13:             On chip, compute $\mathbf{S}_i^{(j)} = \mathbf{Q}_i \mathbf{K}_{k,j}^T \in \mathbb{R}^{B_r \times B_c}$.

14:             On chip, compute $\mathbf{P}_i^{(j)} = \exp(\mathbf{S}_{ij} - L_{i,k}) \in \mathbb{R}^{B_r \times B_c}$.

15:             On chip, compute $\mathbf{dV}_{k,j} \leftarrow \mathbf{dV}_{k,j} + (\boldsymbol{w}_k \mathbf{P}_i^{(j)})^\top \mathbf{dO}_i \in \mathbb{R}^{B_c \times d}$.

16:             On chip, compute $\mathbf{dP}_i^{(j)} = \mathbf{dO}_i \mathbf{V}_j^\top \in \mathbb{R}^{B_r \times B_c}$.

17:             On chip, compute $\boldsymbol{dW}_{i,k} = \mathrm{rowsum}(\mathbf{P}_i^{(j)} \circ \mathbf{dP}_i^{(j)})$.

18:             On chip, compute $\mathbf{dS}_i^{(j)} = \boldsymbol{w}_k \mathbf{P}_i^{(j)} \circ (\mathbf{dP}_i^{(j)} - D_{i,k}) \in \mathbb{R}^{B_r \times B_c}$.

19:             Write $\boldsymbol{dW}_{i,k}$ to HBM.

20:             Load $\mathbf{dQ}_i$ from HBM to SRAM, then on chip, update $\mathbf{dQ}_i \leftarrow \mathbf{dQ}_i + \mathbf{dS}_i^{(j)} \mathbf{K}_j \in \mathbb{R}^{B_r \times d}$, and write back to HBM.

21:             On chip, compute $\mathbf{dK}_{k,j} \leftarrow \mathbf{dK}_{k,j} + \mathbf{dS}_i^{(k,j)^\top} \mathbf{Q}_i \in \mathbb{R}^{B_c \times d}$.

22:         **end for**

23:         Write $\mathbf{dK}_{k,j}, \mathbf{dV}_{k,j}$ to HBM.

24:     **end for**

25: **end for**

26: $\boldsymbol{dW} = \boldsymbol{dW}.\mathrm{sum}(\dim = 0)$

27: Return $\mathbf{dQ}, \mathbf{dK}, \mathbf{dV}, \boldsymbol{dW}$.

## A.3 More case studies

*...An alternate approach is given below in Corollary \ref{cor:infty}.* Along the way we obtain more information about the eigenfunctions, which leads directly to an explicit formula for $u_m(x;\infty)$, see \eqref{conjsum2} and \eqref{Bkmexplicit}. As $\sigma$ increases, the derivatives of $u_m(x;\sigma)$ remain bounded, and so to ensure that the interior condition in \eqref{deltabc} continues to hold, the values $u_m(x_k;\sigma)$ *converges to infinity....*

*...must converge to 0 as $\sigma$ converges to infinity. Our* first corollary of Theorem \ref{thm:main} is that these values converge to $0$ at the same rate for each node $x_k$. \begin{cor} \label{cor:nodes} . *Up to an overall normalization factor, for each $\sigma \geq 0$ and $1 \leq m \leq n-1$, the values of the eigenfunctions...*

*...To obtain the limiting eigenfunctions, which* we denote by \label{def:uminfty} $\nu_m(x;\infty)$ $=$ $\lim_{\sigma \to \infty} u_m(x;\sigma)$, one can use the fact that $\gamma_m(\sigma) \to n\pi$ *for $1 \leq m \leq n$ to obtain...*

*...the eigenvalues $la_m(\sigma)$ for $1 \leq m \leq n$ all converge to $la_n = n^2\pi^2$ as $\sigma$ tends* **to infinity. (Note that this is consistent with our implicit expression for the eigenvalues $la_m(\sigma)$ from Theorem \ref{thm:main} .)** From Corollary \ref{cor:nodes} , this ensures that $u_m(x_k;\sigma)$ converges to zero as $\sigma$ tends to infinity. This means that $u_m(x;\infty)$ (defined in \eqref{def:uminfty} ) is proportional...

Figure 5: In the case above, retrieved top-1 chunk introduces the definition used in the target chunk, while the adjacent retrieved 3rd best chunk and retrieved 4th best chunk both cover the same variants as those appear in the target chunk. Retrieved 2nd best chunk contains the theorem and corollary used in the **query chunk**.

*... denote the projection $\pi : \rpvc \to \cpvc.$* \begin{lemma} \label{lemma:mu circ pi=2n} *The complex and real moment maps for $G^{\mathbf{C}}$ are related by $\mu^* \circ \pi = 2n$* \end{lemma} \begin{proof} *Many of our computations...*

*... $\pi(\omega([v])) = \omega(\pi[v])$* **where $\omega(p)$ denotes the $\omega$-limit set of the negative gradient flow starting from $p$.** \end{prop} \begin{proof} **Applying Lemma** \ref{lemma:mu circ pi=2n} we have $4 < grad||n||^2[v], w_{[v]} >= 4$ ...

Figure 6: In the case above, retrieved top-1 chunk introduces the lemma used in the target chunk.

*... They are not the same: see Section \ref{sec:15}. To establish Theorem \ref{thm:ABn} it suffices to prove it for $B(n)$; the estimate for $A(n)$ then follows from the linear relation $A(n) = \log G_n + B(n)$* (from \eqref{eqn:GABx} ) *combined with the asymptotic estimate for $G(n)$ in \eqref{eqn:logG-asymp}. The main contribution* in the sum $B(n)$ comes from those primes $p$ having $p > \sqrt{n}$, whose key property ...

*...* that exponential sum methods yield alternative unconditional estimates for $A(n,x)$, $B(n,x)$ and $\log G(n,x)$, which are nontrivial when $x = o(n)$, and apply for $x > \sqrt{n}$. These estimates improve on the estimates of our main theorems for certain ranges ...

*...* exponents $\nu_p(G_n)$ as a difference of quantities given by statistics of the base $p$ radix expansion of integers up to $n$ (see Theorem \ref{thm:explicit} ). Summing over $p \leq x$ yields a formula $\log G(n,x) = A(n,x) - B(n,x)$ involving nonnegative arithmetic *functions $A(n,x)$ and $B(n,x)$ ...*

*...* **The implied constant** in the $O$-notation does not depend on $\alpha$. \end{thm} **The limit function $f_B(\alpha)$ is pictured in Figure \ref{fig:B2}. The function lies strictly above the diagonal line** $\beta = (1-\gamma)\alpha$; note that in \eqref{eqn:GABx} in its relation to $\log G(n,x)$ it appears with a negative sign, consistent with $f_G(\alpha)$ ...

Figure 7: In the case above, retrieved top-1 chunk and retrieved 3rd best chunk are adjacent, which mentions the same equation as target chunk. retrieved 2nd best chunk and retrieved 4th best chunk both mention $\log G(n,x)$ , which also appears in target chunk.

## A.4 THE DETAILS FOR THE NIAH TEST

In all evaluations conducted for the NIAH tests, we fine-tune all models using checkpoints derived from PG19, employing the same set of synthetic data. The number of fine-tuning steps is set to one-tenth of the total steps used during pre-training, while all other hyperparameters are kept constant. Examples of the synthetic data utilized for each task are presented in the table 5. Specifically, we pad the input tokens to ensure that the landmark token can be inserted before "is" in the question.

| Task | Example |
|---|---|
| Single NIAH | (essays)... 
 The passkey is: {tokens}. 
 ... 
 What is the passkey? The passkey is {tokens}. |
| 2-hop NIAH with noises | (essays)... 
 DEF {tokens_5}->{tokens_6} ... 
 DEF {tokens_2}->{tokens_3} ... 
 DEF {tokens_4}->{tokens_5} ... 
 DEF {tokens_1}->{tokens_2} ... 
 ... 
 The path from {tokens_1} is: {tokens_2}, {tokens_3} |

Table 5: Task examples for the two NIAH tests.

## A.5 THE DETAILS FOR INFINITELY LONG CONTEXT RETRIEVAL

We pre-train $DRT_{enc\text{-}dec}$ with 200M parameters on MiniPile and store all landmark representations in a Faiss as described in § 3.3 to emulate an infinite context. The trained model has an embedding dimension of 1,024, and an FFN size of 2,816. We train DRT on MiniPile for 20 epochs with 384K tokens per batch. Specifically, we prepare DRT with different settings. $DRT_{w/\ random\ retrieval}$ uses a randomly generated vector for retrieval. $DRT_{w/\ Contriever}$ utilizes a pre-trained retriever with fixed parameters to select top-$k$ relevance chunks, with information still fused via GCA.

## A.6 DISCUSSIONS ABOUT THE $RPT_{CONTRIVER}$ BASELINE

In the original RPT, a reference LM is used to pre-prepare target chunks for each chunk, as discussed in the related works. Compared to using Contriever as the retrieval module, the original method offers stronger causal retrieval capabilities. However, since the code for retriever distillation in the original RPT is not released and the approach is costly and less flexible, we opt to use Contriever instead. $RPT_{contriever}$ can be considered a fusion of RETRO and RPT. It retrieves past chunks in a manner similar to RETRO and integrates the retrieved information in the style of RPT. The retrieval process involves dividing every 64 tokens into a chunk, encoding them with Contriever to obtain chunk representations, and then retrieving past chunks based on cosine similarity with the current chunk.

