# OpenReview forum: "Efficient Long-range Language Modeling with Self-supervised Causal Retrieval"
_ICLR.cc/2025/Conference — Submitted to ICLR 2025_

### Official Review · Reviewer_UH71 · 2024-10-21

**Soundness:** 3
**Presentation:** 3
**Contribution:** 2
**Rating:** 6
**Confidence:** 3

**Summary:**

The paper proposes a novel long-range language model called Differentiable Retrieval-based Transformer (DRT), which introduces a Grouped Cross-Attention (GCA) module to enable joint pre-training of the retriever and causal language model. The main innovation lies in splitting input sequences into chunks and dynamically retrieving relevant past chunks to enhance the prediction of subsequent tokens. The retriever learns how to minimize the auto-regressive loss by retrieving past chunks in a differentiable manner, allowing more efficient training with context lengths up to 64K tokens.
With GCA, the model maintains random-access flexibility during auto-regressive generation while reducing memory and computational costs. Experiments show that DRT outperforms baseline long-range language models by achieving lower perplexity, faster inference speeds, and more efficient memory usage.

**Strengths:**

1. The introduction of Grouped Cross-Attention enables joint optimization of the retriever and language model, solving the issue of fixed retriever parameters in traditional models and making the retriever more adaptive to causal language modeling tasks.
2. The model minimizes memory consumption by performing chunk-based retrieval and leverages hardware optimization to accelerate both training and inference.
3. By offloading historical chunk representations to CPU memory and reloading them as needed, the model significantly reduces GPU memory usage and achieves up to 100 times faster inference compared to existing baselines.

**Weaknesses:**

The method requires pre-training from scratch, as it cannot simply add modules to fine-tune existing models, leading to a substantially higher computational cost.

**Questions:**

1. In Figure 2, the tokens in chunk c7 rely on the relevance score from the LMK in chunk c6 regarding the previous chunks instead of from the token in c7 regarding the previous chunks. Is there any inconsistency?
2. As noted in previous work, there are some problems associated with using PPL as a metric. Do you have any other tasks or metrics that could demonstrate the effectiveness of your method?
3. Could you provide the performance results of your model on long-context benchmarks such as LongBench?

---

> ### Author Response · Authors · 2024-11-17
> **Author Response**
>
> We sincerely appreciate your valuable comments.
>
> **Q1: In Figure 2, the tokens in chunk c7 rely on the relevance score from the LMK in chunk c6 regarding the previous chunks instead of from the token in c7 regarding the previous chunks. Is there any inconsistency?**
>
> This is due to the causal generation process. When generating chunk c7, it is not yet available, so it’s impossible to produce its landmark representation. That is why it's called causal retrieval. While generating the next token, only previous tokens are accessible. Similarly, while generating the next chunk, only previous chunks are accessible.
>
> The main idea of DRT is to use the current chunk (c6) to retrieve past chunks (c1...5) for the next chunk (c7). Tokens in c7 then perform cross-attention with chunks retrieved by c6 to fuse the retrieved information.
>
> For example, consider the NIAH test in General Response 1. Given the current input "What is the passkey? The passkey is: ", we can actually only use the question, rather than the answer, to retrieve relevant information.
>
> More cases can be found in Figure 4 and Figure 7, which show how the model learns to retrieve past symbols based on the current input (in black) for the next chunk prediction (in shallow gray).
>
> **Q2,3: Do you have any other tasks or metrics that could demonstrate the effectiveness of your method?
> Could you provide the performance results of your model on long-context benchmarks such as LongBench?**
>
> Yes, we report new downstream results to show the distinctive properties of DRT in General Response 1.
> Since all models are small and pre-trained from scratch, their performances on evaluation datasets like Longbench and RULER, are generally poor without fine-tuning, making it meaningless to compare with each other. Therefore, we conduct fine-tuning on similar tasks before evaluation. This includes long-text summarization and various needle-in-a-haystack (NIAH) tests.
> We believe the high accuracy with ultra-long contexts on the NIAH test can fully demonstrate the effectiveness of GCA.
>
> **W1. The method requires pre-training from scratch, as it cannot simply add modules to fine-tune existing models, leading to a substantially higher computational cost.**
>
> Indeed. Since the main focus of this work is to propose the new GCA module and the DRT architecture, training from scratch allows for a fair comparison with similar methods and ablation studies.  In fact, the GCA module can be integrated with existing LLMs by incorporating additional GCA modules. However, fine-tuning may pose some challenges and necessitate specific techniques that are beyond the scope of this paper. We plan to explore this further in future work.

---

> > ### Comment · Reviewer_UH71 · 2024-11-23
> > **Response to authors**
> >
> > Thank you for addressing my questions and concerns. I will stand by my score.

---

> > > ### Author Response · Authors · 2024-11-25
> > >
> > > Thank you for your reply!

---

### Official Review · Reviewer_XkMg · 2024-10-28

**Soundness:** 3
**Presentation:** 2
**Contribution:** 3
**Rating:** 5
**Confidence:** 3

**Summary:**

This paper introduces a new approach to enhancing the efficiency and effectiveness of retrieval-based language models through a mechanism called Grouped Cross-Attention (GCA). The authors address the limitations of existing models, which typically rely on separately pre-trained retrievers with fixed parameters that may not optimally align with causal language modeling objectives. The chunk selection part of GCA can be optimized with language modeling loss, and this method can be implemented based on Flash-attention 2, which remarkably reduces the training and inference cost.

**Strengths:**

1. the  proposed GCA method  facilitates the joint pre-training of the retriever and the causal language model in an end-to-end manner. This allows the retriever to dynamically learn to select past chunks.
2. The integration of top-k retrieval and hardware-aware implementations (based on FlashAttention-2) ensures that DRT achieves lower perplexity with reduced training time, outperforming  baselines like Retrieval-Pretrained Transformer and Landmark Attention.
3. During inference, DRT employs memory offloading techniques to manage GPU memory usage efficiently, substantially reducing the memory footprint while maintaining high performance.

**Weaknesses:**

1. The proposed GCA module is difficult to understand because the authors do not provide sufficient motivation for the design of each step. Additionally, there are alternative methods to implement the differentiable operation, but the paper does not include any analysis or comparison of these alternatives. This lack of clarity and justification makes it challenging for the community to follow and reproduce the work.
2. My primary concern is the lack of scalability. Most of the baselines use models with fewer than 200M parameters. The authors mention having access to 8 A100 GPUs, which should enable the training of larger-scale models. I believe this method could benefit from being adopted with a fine-tuning approach, such as CEPE[1], in combination with existing  LLMs like Llama to demonstrate its effectiveness at a larger scale.
3. The authors only provide experiments on language modeling. Evaluating the method across a diverse range of downstream tasks would allow the community to better assess the overall effectiveness and applicability of this work.

**Questions:**

See weaknesses.

---

> ### Author Response · Authors · 2024-11-17
> **Author Response**
>
> We sincerely appreciate your valuable comments.
>
> **W3: Regarding downstream task evaluation.**
>
> Thanks for the suggestion. We show evaluation results on some downstream tasks in General Response 1.
>
> **W2: Concerns regarding the scalability...I believe this method could benefit from being adopted with a fine-tuning approach, such as CEPE[1],....**
>
> Many thanks for your kind suggestion!
>
> We trained all models from scratch to ensure a fair comparison, with results for larger models reported in General Response 3. The key evidence demonstrating the effectiveness of GCA is that even a 133M DRT can significantly outperform a 355M BaseLM on the NIAH test.
>
> Actually, we also considered fine-tuning based on existing LLMs like CEPE.
> However, the components used in CEPE, such as Contriever and cross-attention, are well-established, and therefore its core motivation focuses on how to fine-tune large models to support longer contexts.
> In contrast, our GCA is a newly proposed module that necessitates extensive experiments to explore its effectiveness and compare it with similar methods.
> Directly fine-tuning based on a pre-trained LLM might make it difficult to distinguish whether improvements are due to the additional training steps or the increased number of parameters. Moreover, a decoder with fixed parameters may not learn to retrieve well. How to tackle these challenges is out of the scope of this paper. We will explore this topic in future works.
>
> Moreover, the advantages of DRT are not only in the performance but other distinctive properties as well, as outlined in General Response 2.
>
> **W1(a): Regarding reproducing the work.**
>
> We are preparing to release all code to make sure all results can be reproduced.
>
> **W1(b): The proposed GCA module is difficult to understand ... Additionally, there are alternative methods to implement the differentiable operation, ..**
>
> Regarding alternative methods to implement the differentiable operation, the key issue isn't whether the operation itself is differentiable. As shown in Figure 1(a), the primary issue with conventional approaches is that the outputs of the selection operation are indices rather than values. It's impossible for these indices to propagate gradients, even if the selection operation itself is differentiable. Our core innovation lies in allowing relevance scores to participate in the next token prediction by serving as fusing weights, as illustrated in Figure 1(b). This is further clarified in Lines 45-46 of the updated manuscript.
>
> We are open to discuss other potential options for achieving learnable relevance scores and would appreciate any specific methods you suggest.
>
> Regarding "The proposed GCA module is difficult to understand", sorry for any inconvenience caused to your reading.
> In conventional approaches shown in Figure 1(a), scores are merely used to pick chunk indices, and the top-k retrieved chunks are concatenated for chunked cross-attention. The main idea of GCA is to perform cross-attention on each chunk separately and then fuse them using retrieval scores, as shown in Figure 1(b). This allows scores to receive and propagate gradients from the auto-regressive loss.
>
> During the DRT generation process, the current input uses its landmark representation to retrieve past chunks and selects the top k chunks as the "dynamic context" for the next chunk. Detailed token-to-token cross-attention is then performed on the retrieved chunks for each token generation in the next chunk. In GCA, the scores can receive backpropagated gradients, allowing the model to learn to assign higher scores to chunks that better help predict the next chunk.
>
> We hope the explanation helps to further clarify the GCA module. If there are specific aspects of the description that remain unclear, please let us know so we can refine the manuscript.

---

> > ### Author Response · Authors · 2024-11-22
> >
> > Dear Reviewer XkMg, we reported the results of 355M and 775M models in General Response 3, where the 355M model includes all strong baselines (Landmark, baselm+2layers). The results show that DRT still leads in long contexts and maintains a dominant advantage in the NIAH tests. We sincerely look forward to your reply on whether your concerns have been addressed.

---

> ### Author Response · Authors · 2024-11-25
>
> Regarding the issue of model scalability, we reported a comparison against CEPE 7B. Even without scaling up, our small DRT is powerful enough to beat the 7B model in the needle-in-a-haystack tests. We sincerely look forward to your reply and hope you can reconsider your score.

---

> > ### Comment · Reviewer_XkMg · 2024-11-26
> >
> > It seems that DRT 133M has a supervised finetuning procedure on NIAH tasks to beat CEPE 7B. The claim that a 133M model is powerful enough to beat the 7B model in the needle-in-a-haystack tests may mislead the community. Please correct me if I have misunderstood. If this is truly the case, I suggest the authors comparing the models using a fair setting and avoid making overclaims such as "133M > 7B".

---

> ### Author Response · Authors · 2024-11-26
>
> Thank you for your reply!
>
> Firstly, please note CEPE performs slightly better than DRT with context under 4K. Within its 4K pre-training context window, CEPE can achieve 100% accuracy on the NIAH test, demonstrating it is fully capable of handling the NIAH test. Thus there is no need for fine-tuning.
> Additionally, CEPE has undergone continue-training on long-context datasets that are dozens of times larger than ours, which is quite unfair for DRT too. Meanwhile, in General Response 3, **we also reported the NIAH test results for 775M baselm with fine-tuning under the same setting**, our 133M model also steamrolls the 755M model. All this evidence shows that we have not overclaimed DRT's capabilities in NIAH tests. **Moreover, we only claimed to be powerful enough in the NIAH tests and did not claim that the 133M model outperforms the 7B model in all tasks.**
>
> Furthermore, if simply fine-tuning could achieve high accuracy on the NIAH tests, why no existing LM architectures manage to achieve high accuracy on the single and 2-hop NIAH tests with over 1M context lengths? Could you kindly provide any precedent for this? We argue the reason why LLMs fail to find a needle in a haystack is that the needle's location exceeds the self-attention window. Even with fine-tuning, LLMs can only achieve 100% accuracy within its attention window. However, due to the quadratic complexity of attention and the length generalization issues, the self-attention window size cannot grow infinitely and thus cannot reach 16M.
>
> By using the learnable causal retrieval mechanism (GCA), we can efficiently retrieve the key needle from the entire context, even 16 million tokens, which is how we achieve this result.

---

> ### Author Response · Authors · 2024-11-26
>
> Regarding comparison using a fair setting, **we already provided these results in General Response 1, all models are samely fine-tuned, with fair parameter sizes.**

---

> ### Author Response · Authors · 2024-11-27
>
> For your convenience, we reorganized previous NIAH results (reported in paper Table2 and General Response 1) as follows:
>
> | Models | #params | finetuned | attn_win | 4K | 16K | 64K | 16384K (16M) |
> | - | - | - | - | - | - | - | - |
> | BaseLM | 133M | yes | 768 | 5.83 | 1.06 | 0.0 | - |
> | BaseLM | 330M | yes | 768 | 17.76 | 4.59 | 1.43 | - |
> | BaseLM | 768M | yes | 768 | 18.29 | 4.95 | 4.29 | - |
> | Landmark attn. | 133M | yes | 768 | 99.82 | 97.88 | 0.0 | - |
> | DRT | 133M | yes | 768 | 99.65 | 99.65 | 100.0 | 100.0 |
> | CEPE | 7B | no | 4K | 100.0 | 55.45 | 6.34 | 0.0 |
>
> The difficulty of the NIAH task lies in whether the model can generalize to input lengths longer than those seen in pre-training while making full use of context information. From the table above, it can be seen that all models except CEPE are finetuned, but the performance of all models except DRT decreases as the input length increases. This is because the needle is located outside the attention window, resulting in the inability to directly access this information via self-attention.
>
> Regarding whether CEPE needs finetuning, the CEPE decoder is pre-trained with a 4K context length, achieving 100% accuracy within the 4K range, proving that finetuning is unnecessary. For contexts beyond 4K, CEPE utilizes a retriever to fetch chunks as references. However, unlike DRT, the retriever for CEPE is trained separately and has fixed parameters, thus finetuning is meaningless.
>
> Overall, the NIAH tests are not tasks that can be simply improved by finetuning; they test the model architecture's ability to utilize long-range information. The 2-hop NIAH tests even require the model's long-range reasoning ability, evaluating the fundamental long-range capabilities of various architectures. Merely increasing the parameters or training data cannot solve these intrinsic flaws. One of the main contributions (contribution 2) of this work is we find an effective module to address the long-range random accessibility issue and have demonstrated these abilities in language modeling and NIAH tests.
>
> We hope our clarification addresses your concerns, and we look forward to your further feedback if you still have concerns.

---

> ### Author Response · Authors · 2024-11-28
>
> To further clarify misunderstandings about the NIAH tests:
>
> Our fine-tuning stage is designed to teach models to follow instructions, which can be regarded as instruction-tuning. All our models are tuned with an attention field equivalent to 768 tokens. The base model for CEPE, Llama 2 7B, has already undergone extensive instruction-tuning with a 4K attention window. When evaluating CEPE, we use the last 256 tokens to produce a query embedding, which is then utilized to retrieve 16 chunks from the input, each containing 256 tokens. This process constructs a 4K context for cross-attention. Consequently, if the retriever fails to retrieve the key chunk, the LLM will fail to find the needle. Maybe CEPE can perform better if we expand the additional context to 128K. However,  how about context beyond 128K? The key point is that DRT can achieve near-perfect accuracy within a range of 16M. **There is a similar discussion about the NIAH test in the original CEPE paper (Appendix D), where they mentioned:
> We find that while CEPE LLAMA-2 is able to perfectly retrieve the needle
> in context at the training sequence length (\~10K tokens) and at some longer sequences (\~14K tokens),
> it struggles at other lengths. This is likely due to
> the training/inference discrepancies on the input
> length and we will explore training objectives that
> can mitigate such discrepancies in the future.**
>
> It's important to note that we did not specifically teach DRT how to handle inputs longer than 16K tokens. The context length during instruction-tuning is just 16K. However, after instruction-tuning, DRT, with its 768-token attention field, can achieve perfect accuracy in the single NIAH test with contexts up to 16 million tokens—1,000 times longer than during instruction-tuning. These results are both significant and inspiring.  According to the RULER paper, only a few LLM can achieve perfect NIAH performance at over 128K, let alone 16M context. It's our architecture improvement (GCA) at the fundamental level that makes this result possible.
>
> We would greatly appreciate it if you could give us some feedback!

---

### Official Review · Reviewer_UYQ7 · 2024-11-04

**Soundness:** 2
**Presentation:** 2
**Contribution:** 2
**Rating:** 6
**Confidence:** 3

**Summary:**

This paper proposes Differentiable Retrieval-based Transformers (DRT). Following a previous line of work on retrieval-augmented LMs (e.g. RETRO and RPT), the key novelty of this paper is Grouped Cross Attention – instead of concatenating top k retrieved units, the authors proposed to use retrieval score to fuse outputs derived independently from each unit in the top K. The authors pre-train a DRT model and several baselines (RPT, Landmark Attention, etc.) from scratch and conduct perplexity evaluation on long-range language modeling tasks. Results show that DRT obtains roughly on-par or better performance compared to the baseline, with better performance as the context length increases.

**Strengths:**

* The paper proposes a method to train retrieval-augmented LMs, aiming to improve LM’s capability to handle long context, which is a practical use-case and active research area.
* The paper contains comprehensive experiments against several baselines as well as ablation studies.

**Weaknesses:**

While I appreciate the inclusions of different baselines (RPT, Landmark attention) and the motivation of training RALM in an end-to-end manner,  I am finding it a little bit hard to situate this paper against previous work  and the major benefit of the proposed method:
* It seems like the authors are advocating for the use of GCA, which enables training time efficiency due to end-to-end training. Yet, if I am reading table 1 correctly, it seems like baseLM (+2 layers) achieve lower perplexity on PG19 consistently compared to DRT and also with lower training wall clock time.

* It is also unclear how to isolate the benefit of GCA, given that the baselines have several other different design choices (RPT might be the closest, but it doesn’t use the landmark tokens as this paper, nor grouping the layers) and there is no ablation of using GCA v.s. a different method to train the retrieval module (e.g. the method employed by the RPT model which involves using a reference LM to score the retrieved chunks). It would be helpful to include an ablation study on GCA, besides those that are already included in Table 1.

**Questions:**

* The original RPT paper uses representation from the lower layer to retrieve chunks for upper layers, and train the retriever using a reference model. This paper implements RPT using a contreiver to retrieve chunks, how is that exactly done, i.e. what is the representation for the query and the chunks? And is a fixed contriever used? If so, this seems like a weaker baseline than the originally-proposed RPT, it would be helpful to include a discussion on this.

* The paper mentioned that the proposed method enables “multi-hop retrieval” several times in the paper (e.g. second paragraph in section 3.1), what does this mean and what’s the experiment that demonstrates the ability?

* It seems like the proposed method perform better on the Arxiv-math dataset compared to the PG-19 dataset and it'd be helpful to include analysis to understand the reason. It'd also be helpful to experiment on other domains that are tested in previous work, such as Code [0] and books [1].

[0] https://arxiv.org/pdf/2306.13421
[1] https://arxiv.org/pdf/2402.16617

---

> ### Author Response · Authors · 2024-11-17
> **Author Response (part.1)**
>
> We sincerely appreciate your valuable feedback.
>
> **Q3: It seems like the proposed method perform better on the Arxiv-math dataset compared to the PG-19 dataset and it'd be helpful to include analysis to understand the reason. It'd also be helpful to experiment on other domains that are tested in previous work, such as Code [0] and books [1].**
>
> We appreciate the suggestion that validating on a wider range of datasets would strengthen our findings.
>
> Firstly, although the Code dataset is open source, files are unordered. To fully utilize the long-context modeling capability, the code needs to be concatenated into a long document according to their dependency order. This preprocessing script has not been released by previous work, making comparisons difficult.
> Secondly, the book datasets are somewhat homogeneous compared to PG-19.
> Based on the above reasons, we instead evaluate DRT on multiple downstream long-text tasks akin to RULER for further analysis. For detailed results, please see General Response 1 or Table 2.
>
> Meanwhile, after a new adjustment to architecture, performance on PG-19 and arXiv of our model (Table 1) is further improved when context length exceeds 16K. Please refer to General Response 4 for details.
>
> **W1: it seems like baseLM (+2 layers) achieves lower perplexity on PG19 consistently compared to DRT ...**
>
> In the original manuscript, $DRT_{\text{retrieval} \times G}$ already consistently achieved lower perplexity compared to BaseLM, when the content length exceeded 16K. The DRT you referred to might be $DRT_\text{enc dec}$, which is a weaker version of the standard DRT.  In $DRT_{\text{retrieval}\times G}$, the encoder is applied to the outputs of the lower layers, whereas in $DRT_\text{enc dec}$, the encoder is directly applied to the input embeddings.
>
> **Q1: The original RPT paper uses representation from the lower layer to retrieve chunks for upper layers, ...**
>
> $RPT_\text{Contriever}$ can be considered a fusion of RETRO and RPT. It retrieves past chunks like RETRO and integrates the retrieved information in the style of RPT. The retrieval process involves dividing every 64 tokens into a chunk, encoding them with Contriever to obtain chunk representations, and then retrieving past chunks based on cosine similarity with the current chunk. Consequently, $RPT_\text{Contriever}$ does not require landmark tokens to summarize chunk information, as this task is effectively handled by Contriever.
>
> We acknowledge that this may be a weaker baseline compared to the original RPT, which is why we distinguish it with a subscript. However, the original RPT's code for distilling past chunks from a reference LM is not available, and even if it were, distillation makes the comparison unfair. Moreover, it lacks flexibility for fine-tuning downstream tasks as mentioned in related works Lines 107-110.
> Hence, we used Contriever as an alternative. We have already added an additional section in Appendix A6 to discuss the issue.
>
> **Q2: The paper mentioned that the proposed method enables “multi-hop retrieval” several times in the paper (e.g. second paragraph in section 3.1), what does this mean and what’s the experiment that demonstrates the ability?**
>
> The "multi-hop retrieval" is a distinctive feature when the retriever is learnable. $DRT_{\text{retrieval}\times G}$ denotes retrieval occurring every $\frac{N}{2G}$ layers, repeated $G$ times. For instance, with $N=12$ and $G=2$, retrieval happens every 3 layers.
>
> In the 2-hop NIAH case from General Response 1, given the input "The path from M is," the DRT should first retrieve "DEF M->A" using "M" as a clue (group 6-9). While this initial retrieval might not fully answer the query, the model can identify a new clue "A" between layers 6 and 9. Based on the prior retrieval's information, e.g. the landmark representation output by layer 9, it can then retrieve "DEF A->K" in the second retrieval (group 10-12), allowing it to answer the question. Essentially, it enhances retrieval by learning from token predictions to improve subsequent retrievals based on previous results. For the detailed formal definition, please refer to Equation 1, where $g$ in $h_t^g$ denotes the group index.
>
> In contrast, baselines with fixed retrievers or static chunk concatenation (like RPT) cannot support multi-hop retrieval.
>
> To demonstrate multi-hop retrieval, we compare $DRT_{\text{retrieval}\times 1}$ and $DRT_{\text{retrieval}\times 2}$ in Table 1. More significant results in Table 2 (or General Response 2) show that $DRT_{\text{retrieval}\times 2}$ significantly outperforms $DRT_{\text{retrieval}\times 1}$ in the 2-hop NIAH test, highlighting the multi-hop retrieval capability of $DRT_{\text{retrieval}\times G}$.

---

> ### Author Response · Authors · 2024-11-17
> **Author Response (part. 2)**
>
> **W2: several other different design choices (RPT might be the closest, but it doesn’t use the landmark tokens as this paper, nor grouping the layers) and there is no ablation of using GCA v.s. a different method to train the retrieval module (e.g. the method employed by the RPT model which involves using a reference LM to score the retrieved chunks)**
>
> Thank you for your valuable suggestion.
>
> In Table 4, we present $DRT_\text{w/ Contriever}$, a variant where the retrieval module is replaced with Contriever, allowing us to compare retrievers trained using different methods. This helps determine if the learned retriever performs better than a differently trained one.
>
> We also compare DRT with $RPT_\text{Contriever}$ to assess the effectiveness of GCA against a fixed retriever with Chunked Cross Attention. Both models have nearly identical architectures, except for GCA. RPT does not use landmark tokens since its fixed retriever doesn't require special tokens for chunk representations, as explained in our response to Q1. Additionally, since the retriever is not learnable, performing retrieval for every $G$ layer is pointless as the query representation remains unchanged. Thus, despite lacking landmarks and grouped layers, $RPT_\text{Contriever}$ retains functional modules similar to $DRT_{\text{retrieval}\times 1}$ and serves as an ablation study for GCA.
>
> We did not adopt the RPT method, which utilizes a reference LM to score the relevance of past chunks, for two reasons. Firstly, RPT has not released the code for retriever distillation (https://github.com/OhadRubin/RPT). Secondly, the method is complex and inflexible for both pre-training and post-training, as explained in lines 107-109. Even in the original RPT paper, they did not evaluate their model on downstream tasks.
>
> Moreover, in General Response 1, DRT achieved 100\% accuracy in the NIAH task with 16 million contexts. To our knowledge, no other retrieval-based LMs have achieved such results so far.

---

> > ### Author Response · Authors · 2024-11-22
> >
> > **Regarding the major benefit of the proposed method:**
> >
> > In General Response 2, we outline the advantages of DRT over previous works. Notably, compared to RPT, DRT does not require an external reference LM to guide the retriever, making both pre-training and post-training more flexible. Additionally, the learnable retriever can achieve multi-hop retrieval, unlocking the potential of RALM.

---

> ### Author Response · Authors · 2024-11-30
>
> Dear Reviewer UYQ7:
>
> Thank you once again for your time and for providing your valuable review comments. We are eager to receive your response at your earliest convenience.
>
> As the discussion phase draws to a close, we hope to ensure that all of your concerns have been adequately addressed. To that end, we would like to provide a brief summary of some key points from our rebuttal:
>
> 1. There appears to be a misreading regarding Table 1. We would like to clarify that the perplexity (PPL) of DRT is consistently lower than other baselines when context length exceeds 16K.
>
> 2. The major benefit of DRT: In passkey retrieval (Table 2 in paper or General Response 1), DRT achieves perfect accuracy with **16 million (16384K) contexts**. With a 728 window size and fine-tuning on 16K contexts, it achieves 1000-fold length generalization. This surpasses all known model architectures, including GEMINI 1.5 which scales the attention window to 1M using ring attention. This is significant evidence demonstrating the benefit of the GCA module, extremely long effective context length (>4M) with low training and inference costs.
>
> 3. Compared with RPT, GCA based long-range LMs don't have to rely on an external LLM to distill a retrieval module. This greatly enhances the flexibility and convenience of both pre-training and post-training, contributing to the development of retrieval-based long-range LMs like RPT.
>
> We appreciate your time and consideration. We look forward to your response.

---

> > ### Comment · Reviewer_UYQ7 · 2024-12-02
> >
> > Thank you for your response! I will stand by my score.

---

> > > ### Author Response · Authors · 2024-12-03
> > > **Thanks!**
> > >
> > > Thanks for the response!
> > >
> > > Authors of Submmison 490

---

### Official Review · Reviewer_DDqm · 2024-11-04

**Soundness:** 3
**Presentation:** 3
**Contribution:** 3
**Rating:** 6
**Confidence:** 3

**Summary:**

This paper aims to build a differentiable retrieval-based transformers (DRT) that optimizes the retriever together with generation in an end-to-end manner. It proposes to replace the Chunked Cross Attention with Group Cross Attention that allows the gradient to back-propagate to the retriever so that retriever can be trained as well where in previous work, the retriever is not trained. The evaluation on PG19 and ArXiv-math shows the effectiveness of the proposed method while being much faster than landmark attention.

**Strengths:**

The authors successfully trained a differentiable retrieval-based transformers where adding grouped cross attention shows better perplexity compared to the baseline model.

**Weaknesses:**

1. All the models are trained using very small model size (i.e. 133M), compare to the popular LLMs, e.g. 1B or 7B model size. The improvement could diminish when scaling to larger model sizes and a larger model size should be reported to make the claim more convincing.
2. It is not surprising that the model gets better perplexity by using more compute. Existing SOTA models mostly choose to scale the data size to better utilize the compute. Thus, a more interesting research question could be that given fixed amount of compute, should we allocate the extra compute to train more data or change the architecture to a differentiable retrieval-based transformer.

**Questions:**

1. Will the improvement over baseline model diminish when scaling up the model size?
2. Will the authors open source the codebase? Making it open source might benefit more researchers.

---

> ### Author Response · Authors · 2024-11-17
> **Author Response**
>
> We sincerely appreciate your valuable feedback.
>
> **W2. given a fixed amount of compute, should we allocate the extra compute to train more data or change the architecture to a differentiable retrieval-based transformer.**
>
> We believe this is a highly worthy topic of discussion.
>
> Firstly, it is important to clarify that our advantages do not stem from increased computation. Actually, DRT can be comparable to BaseLM with full attention when both are pre-trained given the same wall-clock pre-training time:
>
> | Models                    | k  | attn win | PG19 valid |
> |---------------------------|----|----------|------------|
> | DRT$_{\text{retrieval}\times1}$ | 8  | 512      | 14.05      |
> | DRT$_{\text{retrieval}\times1}$ | 8  | 512      | 14.02      |
> | BaseLM                    | -  | 16K      | 14.04      |
>
> It can be observed that we still have a slight lead. We fully agree that such a marginal gain may not justify switching architectures.
> However, there are numerous challenges in long-text scenarios for real-world large language models (LLMs), such as increasing inference costs with input length, extrapolation capabilities, and effective context length. A recent study (https://arxiv.org/pdf/2309.14393) reveals that more energy is consumed during inference than during pre-training.
>
> The greater advantages of switching to the DRT architecture lie in multiple aspects. As shown in Table 2 (or refer to General Response 1), DRT has an ultra-long effective context length, far exceeding the content length used in pre-training, and can maintain 100\% accuracy over 16 million tokens in the single-hop NIAH test. Such efficient long-range random accessibility is a crucial property lacking in current models. Additionally, Table 3 shows that DRT offers very efficient inference speed, requiring only a near-constant KV Cache, which addresses the key limitation of the full-attention approach. The case study in Figure 4 further demonstrates that DRT provides stronger interpretability.
>
> In conclusion, the value of switching architectures lies not only in performance improvement but also in the exciting properties it brings. While training on more data or scaling up attention window size may enhance performance, they won't address the aforementioned challenges (inference efficiency, KV cache cost, extrapolation, long-range random accessibility, long-range multi-hop reasoning, etc.) in long-context scenarios. For a simple comparison between different models, please refer to General Response 2.
>
>
> **Q1. Will the improvement over the baseline model diminish when scaling up the model size?**
>
> Given our limited resources, it is challenging to definitively prove or disprove this issue. While improvements may diminish when scaling up, the model's inherent properties remain unchanged. For instance, our new NIAH test shows the ultra-long effective context length of our model.
>
> As mentioned in General Response 2, performance improvement is just one advantage of DRT. Compared to BaseLM, DRT offers significant reductions in inference cost and a much longer effective context length. These fundamental features of the model remain consistent regardless of scale and may play crucial roles in various long-context downstream tasks.
>
> **W1: a larger model size should be reported to make the claim more convincing.**
>
> Thank you for your suggestion.
> In Table 4, we trained and reported results for a larger model (200M).
> Additionally, results for even larger models (355M) are provided in General Response 3.
>
>
> **Q2. Will the authors open source the codebase? Making it open source might benefit more researchers.**
>
> Sure, we are willing to open-source all our codes including the GQA Triton kernel, models, evaluation, and training.

---

> > ### Author Response · Authors · 2024-11-22
> >
> > Dear Reviewer DDqm, we reported the results of 355M and 775M models in General Response 3, where the 355M model includes all strong baselines (Landmark, baselm+2layers). The results show that DRT still leads in long contexts and maintains a dominant advantage in the NIAH tests. We sincerely look forward to your reply on whether your concerns have been addressed.

---

> > > ### Author Response · Authors · 2024-11-25
> > >
> > > Regarding the issue of model scalability, we reported a comparison against CEPE 7B. Even without scaling up, our small DRT is powerful enough to beat the 7B model in the needle-in-a-haystack tests. We sincerely look forward to your reply and hope you can reconsider your score.

---

> > > > ### Author Response · Authors · 2024-11-27
> > > >
> > > > Dear Reviewer DDqm,
> > > >
> > > > Regarding Weakness 1, we have reported the results of larger models in General 3, specifically training a 775M model, which is close to the 1B size you mentioned.
> > > >
> > > > As for Weakness 2, in the NIAH test, Transformers—whether fine-tuned (BaseLM 775M) or scaled up in size (CEPE 7B)—cannot achieve 100% accuracy on inputs that extend far beyond the pre-training attention window. This limitation is inherent to the Transformer architecture and cannot be mitigated by simply adding more training data or parameters. The fact that the 133M DRT can achieve 100% accuracy in a 16 million context NIAH test is significant. All models were fine-tuned on 16K length; only DRT manages to achieve 100% accuracy in the 16M context, while others fail at just 64K length.
> > > >
> > > > This outcome clearly demonstrates that increasing training data and training time alone cannot resolve the issue of long-context modeling; an architectural upgrade is necessary.
> > > >
> > > > To address your concerns, we have invested a significant amount of time and resources. We would greatly appreciate it if you could give us some feedback.

---

> > > > > ### Comment · Reviewer_DDqm · 2024-11-28
> > > > >
> > > > > Thanks for all the experiments and explanations. I think this is a good exploration for the research community and I have increased my score accordingly.

---

> > > > > > ### Author Response · Authors · 2024-11-28
> > > > > >
> > > > > > Thank you for your reply!

---

### Author Response · Authors · 2024-11-17
**General Response (part. 1)**

We sincerely thank all reviewers for their valuable comments.

**1. Evaluation of downstream tasks**

We conduct evaluations on several downstream tasks and have some exciting findings. All models are pre-trained from scratch, then fine-tuned and evaluated on long-context tasks, including needle-in-a-haystack (NIAH) tests like RULER (https://arxiv.org/abs/2404.06654) and long text summarization (https://aclanthology.org/D18-1206/ , https://aclanthology.org/K16-1028/). Notably, DRT excels in summarization tasks and maintains high accuracy in the NIAH tests, even with context lengths exceeding 1 million tokens. We believe GCA significantly contributes to long-range language modeling, as it not only can extrapolate to contexts much longer than those seen during pre-training, but effectively utilizes distant information as well (single NIAH test), and even can perform multi-hop reasoning (2-hop NIAH test).

Summarization:
| Models | xsum:|R-1 | R-2 | R-L | CNN: |R-1 | R-2 | R-L(Cnn)|
|-|-|-|-|-|-|-|-|-|
| BaseLM | | 29.43| 8.04 | 23.26 | | 32.38|12.57|22.61|
| +2layers| |29.74| 8.26 | 23.48 | | 34.14 | 14.09 | 23.60 |
| Landmark Attn. | | 27.98 | 6.96 | 21.99 | | 34.06 | 13.80 | 23.77  |
| DRT$_{\text{retrieval} \times 1}$ | | 30.30 | 8.59 | 23.92 | | 36.27 | 15.88 | 25.08 |
| DRT$_{\text{retrieval} \times 2}$ | | $\textbf{30.39}$ | $\textbf{8.64}$ | $\textbf{23.98}$ | | $\textbf{36.39}$ | $\textbf{15.96}$ | $\textbf{25.15}$ |


Single NIAH test results:
| Models | Retrieval| 1K | 4K | 16K | 64K | 16M |
|-|-|-|-|-|-|-|
| BaseLM(+2 layers)   | -   | 60.43   | 15.37 | 3.89 | 0.0 | -
| Block Recurrent TFM   | -   | 66.80   | 13.96 | 6.01 | 4.29 | -
| RPT$_\text{contriever}$  | fixed   | 42.13   | 11.66 | 4.24 | 2.86 | -
|Landmark Attn.  | adaptive   | $\textbf{99.98}$  | $\textbf{99.82}$ | 97.88 | 0.0 | -
|DRT$_{\text{contriever}\times1}$  | adaptive   | 98.23   | 98.50 | 98.59 | 100.00 | 100.00
|DRT$_{\text{contriever}\times2}$  | adaptive   | 99.69  | 99.65 | $\textbf{99.65}$ | $\textbf{100.00}$ | $\textbf{100.00}$

An example of the single NIAH:
...(essays) ... The passkey is: YOBDC. ...(essays)...
What is the passkey? The passkey is: (Answer: YOBDC)

2-hop NIAH with noises:
| Models | 1K | 4K | 16K | 64K | 4M
|----------|----------|----------|----------|----------|----------|
| BaseLM(+2 layers)   | 16.60   | 5.83   | 1.06 | 0.0 | -
|Landmark Attn.  | $\textbf{90.82}$   | $\textbf{88.35}$ | $\textbf{86.41}$ | 0.0 | -
|DRT$_{\text{contriever}\times1}$  | 41.07  | 33.39 | 39.93 | 38.57 | 34.28
|DRT$_{\text{contriever}\times2}$  | 88.52  | 84.45 | 86.21 | $\textbf{81.43}$ | $\textbf{79.41}$ |

An example of the 2-hop NIAH with noises:
...(essay)... DEF A->K , ..., DEF G->P, ..., DEF M->A, ..., DEF Z->L ..
Q: The path from M is: (Answer: A, K)

To the best of our knowledge, **DRT is the first model to achieve such high accuracy in the single-NIAH test over 16M and the 2-hop NIAH test over 4M, just with an average attention field of 768 tokens.** Such results fully demonstrate the self-supervised causal retrieval could be a promising direction for long-range language modeling.
A detailed experimental analysis is provided in the updated manuscript.

**2. Advantages of the DRT**

We highlight that the benefits of DRT extend beyond efficient training, as it not only achieves lower perplexity but also exhibits several distinctive properties. Below is a comparison table of various long-context models with similar perplexities under similar pre-training quotas:

| Model | Training Complexity | Inference Time Complexity | Inference Space Complexity | Effective Context Length  |
|-|-|-|-|-|
| Transformer (full attention)| O($NL^2$)  | O($NL^2$) | O($NL$) | $1 \times L$ |
| Transformer (sliding window)| O($NL$)  | O($NL$) | O($C$) | $1 \times W$ |
| Recurrent Models    | O($NL$) | O($NL$) | O($C$) | Limited  |
| Landmark Attention  | O($NL^2$) | O($NL+ \frac{NL^2}{S}$) | O($\frac{NL}{S}$) | $\sim 64 \times L$  |
| DRT | O($NL + \frac{L^2}{S^2}$)| O($NL+ \frac{L^2}{S^2}$) | O($\frac{L}{S}$) | $>256 \times L$  |
| DRT$_\text{offload landmarks}$ | O($NL + \frac{L^2}{S^2}$)| O($NL$) | O($C$) | $>256 \times L$  |

N=number of layers, S = chunk size, L = sequence length, W = sliding window size, C = constant.
The effective context length is measured using the NIAH tests, similar to RULER. DRT$_\text{offload landmarks}$ denotes offloading landmark representations to a faiss datastore, thus reducing time and space complexity of inference to O($NL$) and $O(C)$, respectively.

While performance improvements in language modeling might become marginal with growing parameters, DRT retains advantages in inference speed, low KV cache costs, and ultra-long effective context length. These benefits remain consistent even when scaling up.

---

> ### Author Response · Authors · 2024-11-17
> **General Response (part. 2)**
>
> **3. Regarding model scales**
>
> As a foundational study of a new retrieval and attention module, we need to exclude various factors for a fair comparison with existing baselines (Vanilla TFM, RPT, Landmark attn. etc), and the fairest way is to pre-train all of them from scratch.
>
> Although fine-tuning based on existing LLMs seems to be an option, it doesn't provide a fair comparison. The gains can be attributed to more training steps or additional parameters, making it difficult to demonstrate the effectiveness of GCA and compare it with various other baselines included in our paper.
>
> Given the reasons above, and more importantly, our limited GPU resources, we chose to conduct our research on small model scales.
> Nevertheless, our small model has already shown impressive results in the NIAH tests.
> It's strong evidence to demonstrate the effectiveness of GCA.
>
> Despite the challenges on resources, we trained DRT with 355M parameters and compared the BaseLM with 355M and 330M in the following table. BaseLM has 22 decoder-layers and 1024 hidden dimensions, while DRT has 22 decoder layers (lower layer: 10, upper layer: 2 $\times$ 6 / 4 $\times$ 3) with a 2-layer encoder. All models were trained for 60K steps. Here are the results:
>
> |Models  | eval len | attn win. | PG19 valid | PG19 test | arXiv valid | arXiv test|
> |-|-|-|-|-|-|-|
> | BaseLM | 2K | 512 | 12.98 | 12.09 | 3.15 | 3.15 |
> | BaseLM | 2K | 768 | 12.86 | 11.98 | 3.04 | 3.04|
> | BaseLM(+2layers) | 2K | 704 | 12.78 | 11.89  | 3.08  |  3.08 |
> | Landmark Attn. | 2K | 512 | $\textbf{12.52}$ |  $\textbf{11.64}$  | $\textbf{2.91}$ | $\textbf{2.91}$ |
> | DRT$_\text{retrieval}\times 2$ | 2K | 512 | $\textbf{12.52}$ | 11.66 | 2.98 |  2.98 |
> | DRT$_\text{retrieval}\times 4$ | 2K | 512 | 12.79 | 11.92 | 3.05 | 3.05 |
> |-|-|-|-|-|-|-|
> | BaseLM | 16K | 512 | 12.57 | 11.70 | 2.91 | 2.91 |
> | BaseLM | 16K | 768 | 12.40 | 11.55 | 2.76 | 2.76 |
> | BaseLM(+2layers) | 16K | 704 | 12.33 | 11.47  | 2.80 | 2.80 |
> | Landmark Attn. | 16K | 512 | 12.18 | 11.25 | 2.70 | 2.70 |
> | DRT$_\text{retrieval}\times 2$ | 16K | 512 | $\textbf{12.12}$ | $\textbf{11.22}$  |  $\textbf{2.65}$ |  $\textbf{2.65}$ |
> | DRT$_\text{retrieval}\times 4$ | 16K | 512 | 12.33 | 11.48 | 2.73 | 2.73 |
> |-|-|-|-|-|-|-|
> | BaseLM | 32K | 512 | 12.52 | 11.67 | 2.88 | 2.88 |
> | BaseLM | 32K | 768 | 12.35 | 11.52 | 2.74 | 2.73 |
> | BaseLM(+2layers) | 16K | 704 | 12.29 | 11.43  | 2.78 | 2.77  |
> | Landmark Attn. | 32K | 512 | 12.17 | 11.24 | 2.75 | 2.75 |
> | DRT$_\text{retrieval}\times 2$ | 32K | 512 | $\textbf{12.04}$ | $\textbf{11.21}$  | $\textbf{2.62}$ | $\textbf{2.62}$ |
> | DRT$_\text{retrieval}\times 4$ | 32K | 512 | 12.29 | 11.46 | 2.70 | 2.70 |
>
> The single NIAH test:
> | Models | \#params | 1K | 4K | 16K | 64K |
> |-|-|-|-|-|-|
> |BaseLM  | 330M    |70.20 | 17.76 | 4.59 | 1.43 |
> |DRT$_{\text{retrieval}\times 1}$ | 133M  | 98.23 | 98.50 | 98.59 | 100.00 |
>
> We further conduct experiments on 775M models, where BaseLM has 34 decoder-layers and 1280 hidden dimensions, while DRT has 34 decoder layers (lower layer: 16, upper layer: 3 $\times$ 6)
>
> | Models  | eval len | attn win. | PG19 valid | PG19 test |
> |-|-|-|-|-|
> | BaseLM | 2K | 768 | 12.25 | 11.35 |
> | BaseLM(+2layers) | 2K | 725 | 12.22 | 11.32 |
> | DRT$_\text{retrieval}\times 3$ | 2K | 512 | $\textbf{12.18}$ |$\textbf{11.28}$  |
> |-|-|-|-|-|
> | BaseLM | 16K | 768 | 11.80 | 10.93 |
> | BaseLM(+2layers) | 16K | 725 | 11.77 | 10.90 |
> | DRT$_\text{retrieval}\times 3$ | 16K | 512 | $\textbf{11.74}$ | $\textbf{10.85}$ |
> |-|-|-|-|-|
> | BaseLM | 32K | 768 | 11.76 | 10.90 |
> | BaseLM(+2layers) | 32K | 725| 11.73 | 10.87|
> | DRT$_\text{retrieval}\times 3$ | 32K | 512 | $\textbf{11.70}$ | $\textbf{10.84}$ |
>
> The single NIAH test:
> | Models | \#params | 1K | 4K | 16K | 64K |
> |-|-|-|-|-|-|
> |BaseLM  | 775M    | 70.87 | 18.29 | 4.95 | 4.29 |
> |DRT$_\text{retrieval}\times 3$ | 775M  | 100.00 | 100.00 | 100.00 | 100.00 |
>
> According to scaling law, as the parameters increase, the gains diminish, so the gap in perplexity naturally narrows. However, DRT still maintains a lead across all model sizes and has a significant advantage in long-sequence tasks like the NIAH test. We believe these are strong evidence of our model's scalability.
>
> **4. Architecture adjustment**
>
> In the first version of DRT, we used LlamaModel to wrap all lower layers and each upper layer group. LlamaModel (https://github.com/huggingface/transformers/blob/main/src/transformers/models/llama/modeling_llama.py) normalizes the final output as follows:
> ```
> hidden_states = Norm(last_residual + last_hidden_states),
> ```
> This normalization affects the transmission of residuals from lower layers and upper layer groups, impacting our RPT implementation and all DRT models. After removing unnecessary normalizations and pre-training the models again, we observed a further improvement in perplexity. DRT now outperforms all baselines for context lengths over 16K. We've updated the experimental results in Table 1 in the revised manuscript.

---

> ### Author Response · Authors · 2024-11-25
> **A comparison against CEPE 7B**
>
> Although we are not wealthy enough to train a 7B model, we utilized the DRT small model to compare against the CEPE 7B model (https://github.com/princeton-nlp/CEPE) on the needle-in-a-haystack (NIAH) tests for reference. Specifically, we retain the original 4K context length of the decoder and a 4K extra context for cross-attention. If the input exceeds 8K, we use Contriever to retrieve the 16 most relevant chunks to the input, with each chunk containing 256 tokens, thereby forming an additional 4K context.
>
> Here are the results:
>
> ### Single NIAH Test
> | Models | # Params | 4K | 16K | 64K | 16M |
> |-|-|-|-|-|-|
> | DRT$_{\text{retrieval} \times 2}$ | 133M | 99.65 | $\textbf{99.65}$ | $\textbf{100}$ | $\textbf{100}$ |
> | CEPE | 7B | $\textbf{100}$ | 55.45 | 6.34 | 0.00 |
>
> ### 2-Hop NIAH Test
> | Models | # Params | 4K | 16K | 64K | 16M |
> |-|-|-|-|-|-|
> | DRT$_{\text{retrieval} \times 2}$ | 133M | 84.45% | $\textbf{86.21%}$ | $\textbf{81.43%}$ | $\textbf{79.41%}$ |
> | CEPE | 7B | $\textbf{92.32}$ | 16.48 | 0.0 | 0.0% |
>
> As General Response 3 indicates, DRT consistently excels in fair comparisons on language modeling tasks. In tasks designed for long contexts, such as the NIAH tests, our 133M DRT is already powerful enough to outperform CEPE 7B,  an LLM specially designed for long texts. This demonstrates the power of self-adaptive causal retrieval and the potential of the DRT architecture. We hope that the exploration of fundamental language model architecture is not limited to teams with extensive GPU resources.

---

### Meta-Review · Area_Chair_1VQG · 2024-12-22

**Metareview:**

While DRT presents an interesting approach to long-context modeling through Grouped Cross-Attention and joint training of retriever and language model, the paper's experimental evaluation falls significantly short of the standards. The most critical weakness is the limited scope of experiments: models are only trained at a relative small scale, and evaluations are restricted to basic language modeling metrics without meaningful real-world benchmarks or comprehensive comparisons to state-of-the-art long-context models. The lack of experiments on larger model scales makes it difficult to assess whether the benefits would translate to practically relevant scenarios. Additionally, the baseline comparisons are incomplete. The absence of downstream task evaluations and limited ablation studies further weaken the empirical validation. While the technical approach shows promise, the insufficient experimental rigor and lack of comprehensive benchmarking prevent a clear assessment of the method's practical value, making it fall below the bar for ICLR acceptance.

**Additional Comments On Reviewer Discussion:**

I have read the messages in the discussion period and my opinion has been summarized as in the metareview above. I considered these points in my recommendation.

---

### Decision · Program_Chairs · 2025-01-22

Reject